# The potential of a universal influenza virus-like particle vaccine expressing a chimeric cytokine

Kuniaki Nerome[1], Toshifumi Imagawa[1,2], Shigeo Sugita[3], Youta Arasaki[1], Kenichi Maegawa[1], Kazunori Kawasaki[4], Tsuyoshi Tanaka[5], Shinji Watanabe[6], Hidekazu Nishimura[7], Tetsuro Suzuki[2], Kazumichi Kuroda[8], Isao Kosugi[9], Zenta Kajiura[10]

**The efficacy of the current influenza vaccines is frequently reduced because of antigenic drift, a trade-off of developing improved vaccines with broad cross-protective activity against influenza A viruses. In this study, we have successfully constructed a chimeric cytokine (CC) comprising the M2 protein, influenza A neuraminidase stalk, and interleukin-12. We produced virus-like particles (VLPs) containing CC and influenza hemagglutinin (HA) proteins using a baculovirus system in Eri silkworm pupae. The protective efficacy of the CCHA-VLP vaccine was evaluated in mice. The CCFkH5HA-VLP vaccine increased the survival rates of BALB/c mice, infected with a lethal dose of PRH1 and HKH5 viruses, to 80% and 100%, respectively. The results suggested that CCHA-VLP successfully induced potent cross-reactive protective immunity against infection with homologous and heterologous subtypes of the influenza A virus. This is the first study to design a CC-containing HA-VLP vaccine and validate its protective efficacy.**

## Introduction

The efficacy of the current vaccines against a seasonal influenza virus depends on the influenza season, with 40–80% efficacy reported for healthy adults (Neumann et al, 2020). The reduced efficacy is primarily attributed to the mismatched antigenicity between seasonal epidemic influenza strains and the quantity of antigens in the vaccine. The hemagglutinin (HA) protein, which serves as the major antigen of the influenza vaccine, constantly changes the amino acid sequence because of the high error rate of the viral polymerase and the selective pressure applied by the hosts' circulating antibodies (human herd immunity). This antigenic

drift process causes a vaccine mismatch. Meanwhile, antigenic shift, which can result in the emergence of a pandemic influenza strain derived from a non-human influenza virus strain, can also interfere with vaccine coverage. To overcome these challenges, novel vaccine development strategies are required (Rappazzo et al, 2016; Yao et al, 2019).

Recently, several groups have been working on developing a universal influenza vaccine that can protect humans from various influenza virus subtypes. For example, one study reported a new vaccine strategy that targets the epitope conserved among subtypes in the stem region of the influenza HA protein (Adachi et al, 2019). Specifically, they targeted the long α-helix epitope, which is only exposed in acidic conditions (pH 5.0), as a potential universal vaccine candidate. As another approach, the influenza matrix 2 (M2) protein is also one of the important targets (Turley et al, 2011; Kolpe et al, 2017; Jang & Seong, 2019). In fact, because the M2 protein has a highly conserved amino acid sequence among subtypes of the influenza A virus, the ectodomain of the M2 (M2e) vaccine was reported to have a reactive high protective activity against heterotypic influenza A virus strains. Furthermore, the M2e vaccine induced CD4⁺ and CD8⁺ T-cell responses, suggesting the viral titer would be reduced by antibody-dependent cellular cytotoxicity and IFN-γ production.

The vaccine-induced cellular immunity represents another critical aspect of effective influenza vaccine development. Muramyl dipeptide liposome vaccine, first developed in 1990 and recently renamed as the "VLP vaccine," is another type of influenza subunit vaccine (Iinuma et al, 1995). This vaccine elicited a high HI titer (>1,400) in humans and an apparent cellular immunity in mice (Iinuma et al, 1995; Nerome et al, 2019). Hence, when considering the development of a universal influenza vaccine, the induction of effective cellular immunity is necessary. For instance, NK cells are activated by IFN-α, IFN-β, and IFN-γ, and IL-12, IL-15, and IL-18. Among these, IL-12 and IL-18 specifically activate NK cells (Kobayashi et al, 1989). NK cells and IFN-γ enhancing B-cell

---

[1]Nerome Institute of Biological Resources, Nago, Japan   [2]Department of Microbiology and Immunology, Hamamatsu University School of Medicine, Hamamatsu, Japan   [3]Equine Research Institute, Japan Racing Association, Shimotsuke, Japan   [4]Biomedical Research Institute, National Institute of Advanced Industrial Science and Technology, Osaka, Japan   [5]TORAY Industries, Inc., Iyo, Japan   [6]Research Center for Influenza and Respiratory Viruses, National Institute of Infectious Diseases, Tokyo, Japan   [7]Virus Research Center, Clinical Research Division, Sendai Medical Center, National Hospital Organization, Sendai, Japan   [8]Division of Gastroenterology and Hepatology, Nihon University School of Medicine, Tokyo, Japan   [9]Department of Regenerative and Infectious Pathology, Hamamatsu University School of Medicine, Hamamatsu, Japan   [10]Division of Applied Biology, Facility of Textile Science and Technology, Shinshu University, Ueda, Japan

Correspondence: rnerome_ibr@train.ocn.ne.jp

responses are important for antiviral activity. Previous studies have reported that cytokine treatment is an effective regimen against several communicable diseases (Fujioka et al, 1999; Jiang et al, 1999; Khan et al, 2014; Gai et al, 2017).

We have recently developed an avian influenza H5- and H7-subtype VLP vaccine in a silkworm system (Maegawa et al, 2018). In our previous study, to enhance vaccine immunogenicity based on the series of the above evidence of cytokine usability, IL-12 was incorporated into the influenza HA vaccine. IL-12–containing H5 and H7 VLP vaccines had elevated protective efficacy against a distinct subtype of the influenza A virus; however, this efficacy was still lower than expected (Maegawa et al, 2020). We attempt to address this issue in this study. Specifically, we further created a chimeric cytokine (CC) consisting of IL-12, M2 protein, and headless neur-aminidase (NA) of the influenza A virus. This immune modulator CC protein was co-presented with H1, H5, or H7 antigens to produce a CC-containing HA-VLP (CCHA-VLP) vaccine. Furthermore, we have evaluated the protective efficacy of these CCHA-VLP vaccines against various types of the influenza A virus on a mouse model.

# Results

## Expression of CC and HA proteins and exposure on the surface of Sf9 cells

To investigate the expression of CC, FukushimaH5 HA (FkH5), and AnhuiH7 HA (AnH7) proteins in the insect cells, *Spodoptera frugi-perda* cells (Sf9) were infected with CC-*Autographa californica* nuclear polyhedrosis virus (AcNPV), FkH5-AcNPV, and AnH7-AcNPV and detected with monoclonal or polyclonal antibodies. A series of reaction cells were stained with fluorescent antibodies. Fig 1A and B shows CC-AcNPV–infected Sf9 cells stained in green and red with the anti-M2 and IL-12 antibodies, respectively, indicating the ex-pression of M2 and IL-12 in Sf9 cells. Because commercially available anti-NA antibodies did not work in this experiment, probably because only a part of the NA protein was involved, the FA test with anti-Aichi/2/68 (H3N2) virus polyclonal mouse antiserum was performed. As shown in Fig 1C, cells were positively stained with the antibody. Next, the CC-AcNPV + FkH5-AcNPV–infected cells and the CC-AcNPV + AnH7-AcNPV–infected cells were prepared and the co-expression of HA and IL-12 was partially detected (Fig 1D and E). Although immunofluo-rescence verified the co-expression of the recombinant proteins, this alone did not confirm the exposure of the proteins on the surface of infected cells. To confirm the transport of HA proteins to the cell surface, we performed a hemadsorption assay. Fig 1F shows the negative result for Sf9 cells infected with CC-AcNPV as these cells express the CC protein without HA. In contrast, Sf9 cells co-infected with CC-AcNPV and FkH5- or AnH7-AcNPV show typical hemad-sorption profiles where chicken erythrocytes adsorbed to the co-infected cells (Fig 1G and H).

## Confirmation of the expression of CC and HA proteins by Western blot analysis

To confirm the expression of VLP components in silkworm, VLP samples produced by recombinant Ac-NPV–infected silkworm

pupae were analyzed using Western blot. The band over the 100-kD marker was detected with anti-M2 and anti-IL-12 p70 antibodies in the CC-VLP sample (Fig 2A and B). Similarly, bands below the 75-kD marker were detected with anti-influenza H1, H5, and H7 antibodies in CC Iowa (IA) H1-, CCFkH5-, and CCAnH7-VLP samples (Fig 2C–E). The band of CCAnH7-VLP was relatively weaker than other two VLPs.

## Morphological analysis of the CCHA-VLP structure

Purified VLP samples, including FkH5-, CC-, and CCFkH5-VLPs, were negatively stained with 1% phosphotungstic acid (pH 7.0) and observed with transmission electron microscopy (TEM). First, the surface of the FkH5-VLP was found to be covered with a spike structure (indicated by black arrows, Fig 3A), which resembles the 14-nm spike covering the surface of the H5N1 influenza virus in the reference image (Fig 3B), and is considered to be a spike of HA. Next, spikes were also observed lining the surface of the CC-VLP, which were ~11 nm shorter than the HA spikes (indicated by white arrows, Fig 3C). Fig 3D also shows spikes of 11 nm on the upper surface of the particles in the image, and many of the surfaces were covered with some kind of protein, although most were not clear. Because these VLPs were made from single infection of CC-AcNPV, the spike of ~11 nm in length is likely to be a CC protein. Then, when we observed CCFkH5-VLPs produced of co-infection of FkH5-AcNPV and CC-AcNPV, we noticed a part covered by a 14-nm HA-like spike and an area covered by some structures shorter than HA-like spikes (Fig 3E–G). Most of the detected particles were ~150–200 nm in diameter. Considering the observed structure of CC-VLP, it is likely that CC protein is expressed in the areas not covered by the HA-like spike, and based on the observations of the 40 fields of view that included VLPs, these complex VLPs were the major components of the CCFkH5-VLPs.

## Immune reaction of the CCHA-VLP vaccine based on hemagglutination inhibition (HI) and plaque inhibition tests

At first, HI titers were measured with mice immunized with CCFkH5-VLP or CCAnH7-VLP vaccines (Table 1, up). CCHA-VLPs successfully induced HI antibody against homologous types of challenged vi-ruses; namely, CCFkH5-VLP–immunized mouse serum showed 256 HI titer against the HKH5 virus and CCAnH7-VLP–immunized serum showed 512 HI titer against AnH7 virus. Moreover, after the im-munization with CCFkH5- and CCAnH7-VLP vaccines, the subse-quent challenge infection with A/Puerto Rico/8/34 (H1N1; PRH1), RG-A/Barn Swallow/Hong Kong/1161/2010-A/PR/8/34 H5N1 [R] (6 + 2) (HKH5), and RG-A/Anhui/2013-A/PR/8/34 H7N9 [R] (6 + 2) (AnH7) viruses triggered the production of homologous HI antibody, showing 1,024 to PRH1, 256 to HKH5, and 1,024 to AnH7 viruses. However, HI activity was not detected against the HA genotype different from immunization and virus challenge. Next, to evaluate inducing neutralizing antibodies, a plaque reduction test was performed using mouse sera immunized with CC-VLP, CCFkH5-VLP, or CCAnH7-VLP (Table 1, down). The CC-VLP–immunized mouse serum did not show any neutralizing activity against all challenged viruses, PRH1, HKH5, and AnH7, and CCHA-VLP–immunized mouse sera showed positive neutralizing activity against only homologous

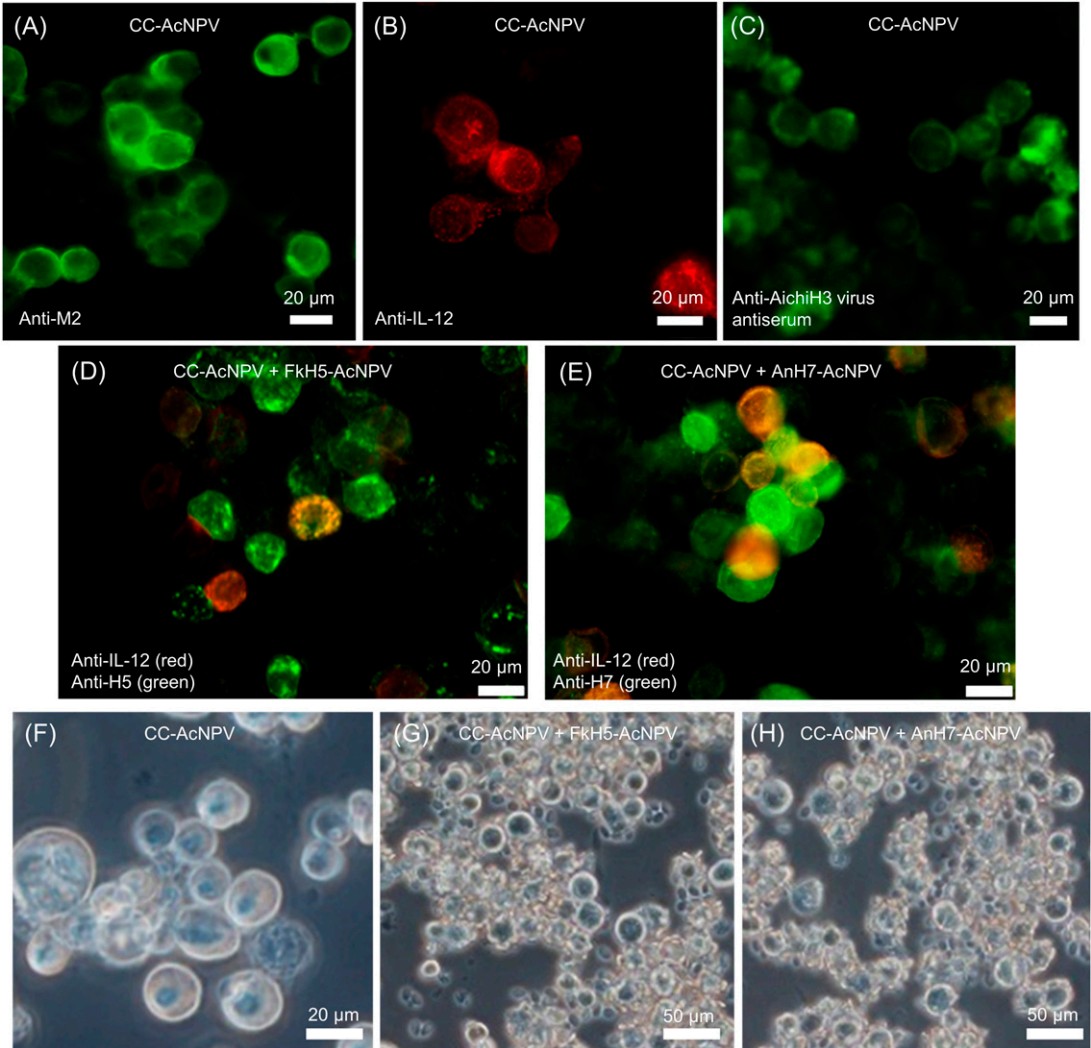

**Figure 1.  Co-expression of CC and HA proteins in Sf9 cells and Eri silkworm pupae co-infected with CC-AcNPV and HA-AcNPV.**
**(A, B, C, D, E, F, G, H)** Sf9 cells were infected with CC-AcNPV (A, B, C, F), or co-infected with CC-AcNPV and either FkH5-AcNPV (D, G) or AnH7-AcNPV (E, H). **(A, B, C, D, E)** Cells were fixed, permeabilized, and stained with anti-M2 (A), anti-IL-12 antibody (B, D, E), anti-Aichi/2/68 (H3N2) virus serum (C), anti-H5 serum (D), and anti-H7 serum (E), followed by fluorescent secondary antibody staining. Anti-IL-12 antibody was stained by Alexa Fluor 555–conjugated antibody, and anti-HAs, anti-M2 antibodies, and anti-Aichi virus serum were stained by Alexa Fluor 488–conjugated antibody. **(F, G, H)** Infected Sf9 cells were also analyzed by the hemadsorption test to confirm the expression of recombinant HA proteins on the surfaces of co-infected Sf9 cells (F, G, H). The CC-AcNPV–infected Sf9 cells were included as a negative control.

types of influenza virus, indicating $10^{3.2}$ and $10^{4.2}$ plaque neutralizing titers against HKH5 and AnH7 viruses, respectively.

### Induction of anti-M2 IgG and IFN-γ–producing cells by CCFkH5HA-VLP vaccine in BALB/c mouse

To evaluate the immunogenicity of M2 protein in the CC molecule, anti-M2 antibody in mice immunized with CCFkH5-VLP was evaluated as relative $OD_{450}$ by ELISA. Anti-M2 antibody was significantly induced in CCFkH5HA-VLP immunized mice, compared with unvaccinated mice (relative $OD_{450}$ = 2.25, $P$ = 0.0003; Fig 4A). Moreover, the number of IFN-γ–producing cells in splenocytes was determined by ELISpot assay to evaluate the induction of cellular immunity by the CCHA-VLP vaccine. As shown in Fig 4B, CCFkH5-VLP–immunized mice had significantly higher IFN-γ–producing cells

per million splenocytes than unvaccinated control mice (mean 19.0 or 39.8 versus 2.2, $P$ = 0.02 or 0.002). The difference between experiments 1 and 2 was the amount of CCFkH5-VLP for immunization: ~2,000 HA/mouse in experiment 1 and 1,000 HA/mouse in experiment 2.

### Cross-protection of CCHA-VLP vaccines against different subtypes of the influenza A virus evaluated by virus replication in the lung

We evaluated the protective efficacy of three CCHA-VLP vaccines, including CCIAH1-, CCFkH5-, and CCAnH7-VLP, against PRH1 and HKH5 viruses in ddY mice (Fig 5A). Briefly, the unvaccinated control mice were treated with PBS and infected with the PRH1 virus. This mouse group showed 40% survival; one mouse died on day 6 and

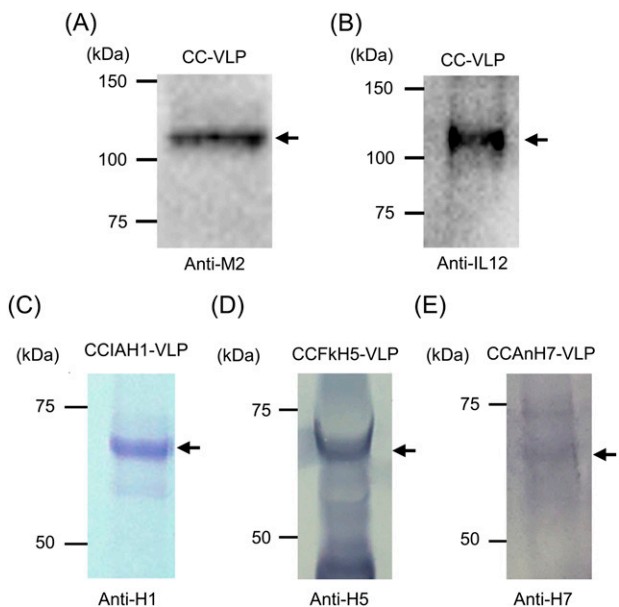

**Figure 2.  Western blot analysis of CC-VLP and CCHA-VLP vaccines.**
**(A, B, C, D, E)** Eri silkworms were infected with recombinant AcNPVs, and the expression of CC-VLP (A, B) and CCHA-VLPs (C, D, E) was confirmed by Western blotting. **(A, B)** CC-VLP was analyzed using anti-M2 antibody (A) and anti-IL-12 antibody (B). The calculated molecular size of CC is 79.6 kD. **(C, D, E)** CCHA-VLPs were analyzed using anti-H1 antibody (C), anti-FkH5 antiserum (D), and anti-AnH7 antiserum (E). **(A, B, C, D, E)** Black arrows indicate bands regarded as CC (A, B) and HA proteins (C, D, E). Uncropped images are shown as source data for figure. Source data are available for this figure.

two mice on day 8 post-infection (Fig 5B). The lungs of the dead mice were pooled, and the plaque titer of the PRH1 virus was examined. As shown in Fig 5F, the plaque titer was $\log_{10}$ 5.1 PFU/ml. Similarly, the challenge infection with the HKH5 virus in unimmunized mice also resulted in 40% survival (Fig 5B), and the plaque titer in the dead mouse lungs was $\log_{10}$ 5.6 (Fig 5G). The mouse groups vaccinated with CCIAH1-VLP (Fig 5C), CCFkH5-VLP (Fig 5D), and CCAnH7-VLP (Fig 5E) vaccines showed a sharp contrast to their protective results, when compared to the control groups treated with PBS. For example, CCIAH1-VLP–immunized mice infected with PRH1 or HKH5 viruses survived for 10 d, and the plaque titer in the pooled survived mouse lungs was less than the detection limit ($\log_{10}$ 1.7 PFU/ml) (Fig 5C, H, I). A similar phenomenon was observed in the CCFkH5-VLP– or CCAnH7-VLP–immunized mice (Fig 5D and E). In particular, all immunized mice with CCFkH5- or CCAnH7-VLP survived for 10 d under PRH1 and HKH5 virus challenge infections, and the plaque titers of those mice were less than the detection limit (Fig 5J–M).

## Protective efficacy of VLP vaccines based on the survival rate of tested ddY mice

The protective activity of three CCHA-VLP vaccines against PRH1 and HKH5 influenza A viruses was directly evaluated based on the survival rate of ddY mice for 10 d after the virus challenge (Fig 6A). PBS-immunized mice challenged with PRH1 and HKH5 viruses showed 20% and 60% survival, respectively (Fig 6B). At first, IL-

12–linked FkH5 (IL-12+FkH5 HA)-VLP, which was included in the previous study, was evaluated as a reference. The survival rate of PRH1-infected mice immunized with the IL-12+FkH5-VLP vaccine without NA and M2 proteins was 60%, showing a low protective activity against heterotypic virus. In contrast, 100% survival was observed in HKH5 virus–infected mice (Fig 6C). Next, the CC-VLP vaccine, which lacks HA antigen, exhibited a minimal protective effect of 60% and 40% against the PRH1 and HKH5 viruses, respectively (Fig 6D). Meanwhile, CCIAH1-VLP, CCFkH5-VLP, and CCAnH7-VLP vaccines exhibited a markedly increased protective activity (Fig 6E–G), showing a 100% survival rate against PRH1 and HKH5 virus challenges, except for CCFkH5-VLP versus PRH1 virus (80%).

## Histopathological examination of mice immunized with and without CC vaccines

The lung tissues obtained from mice immunized with VLP vaccines and challenge-infected with PRH1 (Fig 7) or HKH5 (Fig 8) were histologically examined by staining with hematoxylin and eosin and by immunochemical staining with anti-myeloperoxidase (MPO) and anti-ionized calcium-binding adapter molecule 1 (Iba1) antibodies. MPO and Iba1 were markers of neutrophil and macrophage, respectively, and were used to indicate signs of inflammation. The schedule of lung tissues is shown in Figs 7A and 8A. A dead mouse was selected from the unvaccinated mice, and a surviving mouse at 10 d post-infection was selected from CCFkH5-VLP–immunized mice for this analysis.

No indicators of inflammation, such as immunocyte migration or bleeding into alveolar spaces, were observed in non-infected control mouse lung tissue (Fig 7B–D). Unvaccinated and challenge-infected mice showed evident histopathological damage in the pulmonary alveoli compared with the non-infected control mouse tissue (PRH1: Fig 7E; HKH5: Fig 8B). In contrast, mice immunized with CC-VLP, FkH5-VLP, or CCFkH5-VLP vaccines did not exhibit severe lung inflammation despite challenge infection (Figs 7H, K, and N and 8E, H, and K). Conversely, marked immunocyte infiltration was observed in the lungs of challenge-infected unvaccinated mice, as evidenced by MPO and Iba1 histological stainings (Figs 7F and G and 8C and D). A significant infiltration of immunocytes was not observed in CC-VLP or CCHA-VLP vaccine–immunized mice or in the healthy non-infected mice (Figs 7C, D, I, J, L, M, O, and P and 8F, G, I, J, L, and M).

## Protective efficacy of the CCFkH5-VLP vaccine against six subtypes of the influenza A virus

The CCFkH5-VLP vaccine was further evaluated with BALB/c mouse against PRH1, A/Kumamoto/1/67 (KumaH2), A/Aichi/2/68 (AichiH3), HKH5, AnH7, and A/swine/Hong Kong/10/98 (HKH9) influenza A viruses (Fig 9A). At first, in a challenge with high-virulence viruses, PRH1 and HKH5, all unvaccinated mice died within 10 d post-infection. However, the survival rates of vaccinated mice were 80% in the PRH1 challenge and 100% in the HKH5 challenge (Fig 9B and C). In contrast, in low- and moderate-virulence virus challenges in unvaccinated mice, all mice survived in KmH2 and HKH9 challenges, and 80% of mice survived in AichiH3 and AnH7 challenges,

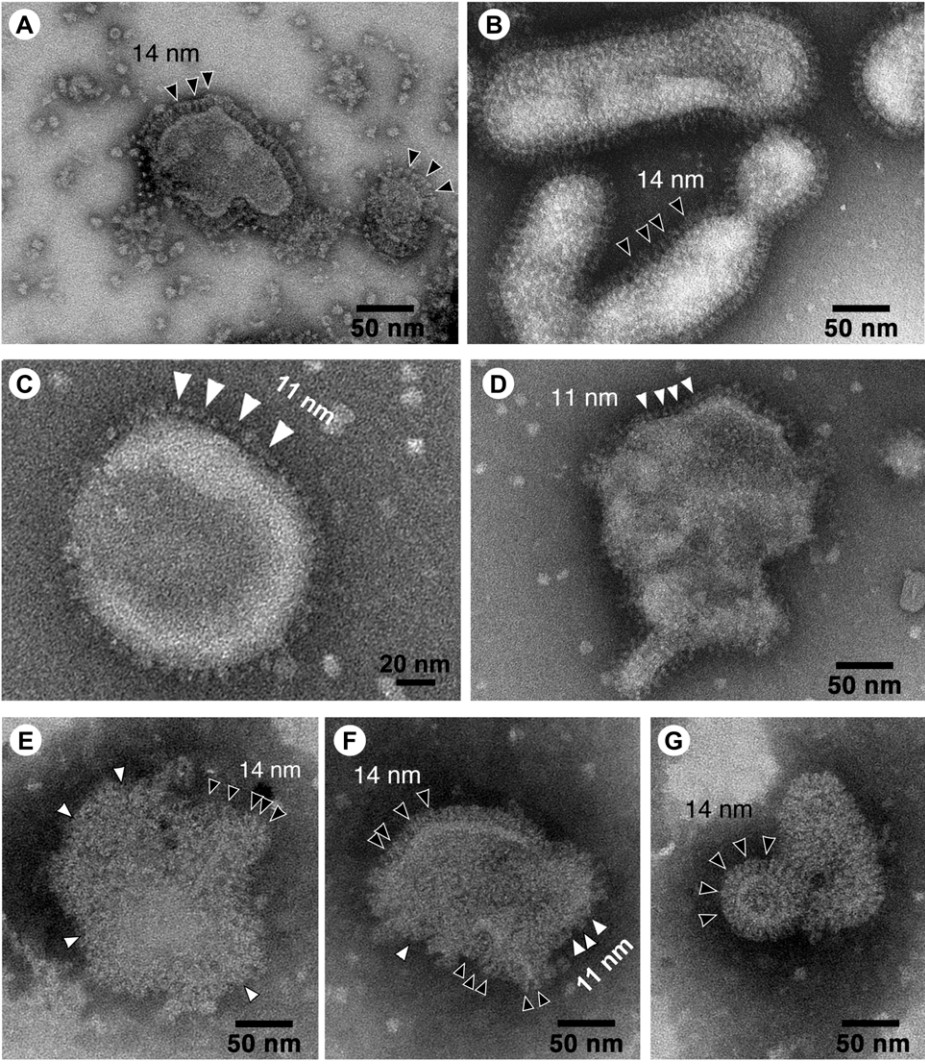

**Figure 3. Morphology of CCH5-VLP produced in silkworm pupae.**
Structure of CCFkH5-VLP, CC-VLP, and FkH5-VLP produced in Eri silkworm pupae was observed using negative staining TEM. Native influenza H5N1 virus produced in fertile hemi-egg.
**(A, B, C, D)** TEM images of FkH5-VLP (A), H5N1 influenza virus (A/Hong Kong/157/1997[H5N1]) (B), which was described in a previous report (Nerome et al, 2015), and CC-VLP (C, D) are shown. White arrows and black arrows indicate 11- and 14-nm projections, respectively. **(E, F, G)** Representative images of CCFkH5-VLP showing both shorter (11 nm, indicated by white arrows) and longer (14 nm, indicated by black arrows) influenza virion-like projections on the surfaces of the particles.

whereas all mice survived in the vaccinated groups (Fig 9D–G). Because it was difficult to compare vaccine efficacy by mortality in low-virulence virus challenge, it was compared by the suppression of viral growth in mouse lungs. Lung viral titers in unvaccinated groups at 4 d post-infection were $10^3$ to $10^4$ PFU/lung in all viruses. In contrast, viral growth in mouse lungs at 10 d post-infection was not detected in vaccinated groups.

## Discussion

In this study, we produced CC comprising the influenza M2, NA stalk, and IL-12 molecules by the baculovirus vector system using Eri silkworm pupae, and evaluated the induction of immune reaction and the protective activity of the influenza HA protein combined with the CCHA-VLP vaccine against several types of influenza A viruses.

The expression of CC and HA proteins was confirmed by immunofluorescence and Western blot analyses. Regarding the expression of VLPs, the Western blot of CC showed a band over 100 kD in molecular size, although the calculated molecular size of CC was ~79.6 kD. This discrepancy was probably due to the sugar chain linkage, as it is estimated that the amino acid sequence of CC includes 10 N-glycan–binding sequences. Next, the result of immunofluorescence analysis with sf9 cells showed that the single infection of either CC-AcNPV or HA-AcNPV was not negligible in the cells co-infected with CC-AcNPV and HA-AcNPV. This may be because the titer of each AcNPV was not considered in this experiment, and thus, the number of infected cells in each AcNPV differs. Because the procedure to purify only hybrid CCHA-VLP was not performed in this study, although it was found that CCHA hybrid VLP was the main component in the sample by TEM observation, it should be considered that it may contain three types of VLPs, CC-VLP, HA-VLP, and CCHA-VLP. However, it is unclear whether the co-existence of CC and HA on the surface of the same VLP particle is important for immunogenicity or not, which thus should be the focus of future studies.

The recent spread of highly pathogenic avian influenza viruses in chickens and humans has increased the need to expedite and improve vaccine development technologies. To address this request, we first established influenza HA-VLP vaccines co-presented

**Table 1. Antibody production in mice immunized with CCHA-VLP vaccines evaluated by the HI test (up) and the plaque reduction neutralization assay (down).**

| | | HI titer to viruses | | | |
|---|---|---|---|---|---|
| **Vaccine** | **Challenge virus** | **PRH1 (H1N1)** | **HKH5 (H5N1)** | **AnH7 (H7N9)** | **HKH9 (H9N2)** |
| CCFkH5HA-VLP | PRH1 | 1,024 | 256 | [a] | ND |
| CCFkH5HA-VLP | HKH5 | — | 256 | — | ND |
| CCFkH5HA-VLP | AnH7 | — | 256 | 1,024 | ND |
| CCAnH7HA-VLP | PRH1 | 1,024 | — | 512 | ND |
| CCAnH7HA-VLP | HKH5 | — | 256 | 512 | ND |
| CCAnH7HA-VLP | AnH7 | — | — | 512 | ND |
| | | Plaque inhibition titer to viruses | | | |
| Vaccine | Challenge virus | PRH1 (H1N1) | HKH5 (H5N1) | AnH7 (H7N9) | HKH9 (H9N2) |
| CC-VLP | | [b] | < | < | < |
| CCFkH5HA-VLP | | < | $10^{3.2}$ | < | < |
| CCAnH7HA-VLP | | < | < | $10^{4.2}$ | < |

[a]HI titer less than 32. ND, not done.
[b]Plaque neutralization titer less than $10^{1}$.

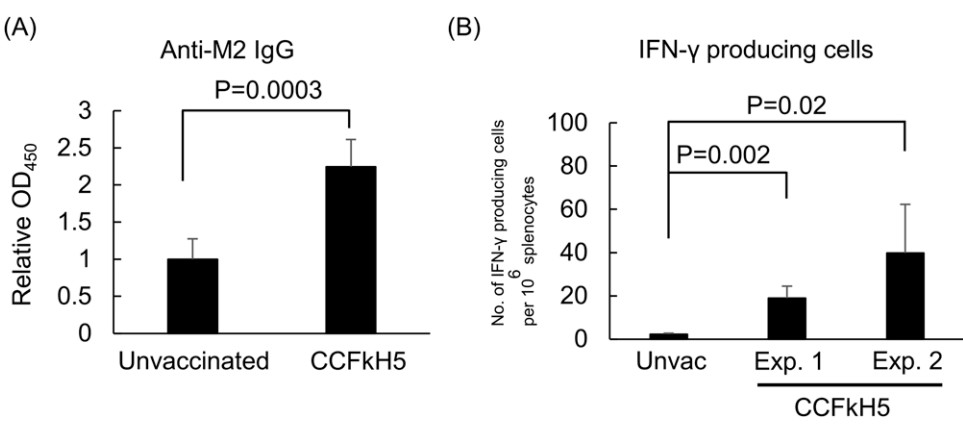

(A) Anti-M2 IgG
P=0.0003

(B) IFN-γ producing cells
P=0.02
P=0.002

**Figure 4. Induction of anti-M2 antibody and IFN-γ–producing cells by the CCFkH5HA-VLP vaccine.**
The CCFkH5HA-VLP was immunized twice in a 2-wk interval, and serum and spleen were collected 2 wk after booster immunization. Anti-M2 antibody and IFN-γ–producing cells in splenocytes were detected with unvaccinated (n = 5) and CCFkH5HA-VLP–immunized (n = 5) mouse groups. Statistical comparison was performed by a t test or Welch's test. **(A)** Anti-M2 antibody was determined as relative $OD_{450}$ by ELISA. The amount of VLP for immunization was 1,000 HA/mouse. **(B)** IFN-γ–producing cells were determined by mouse IFN-γ ELISpot assay. "Unvac" indicates the unvaccinated mouse group. The CCFkH5-VLP–immunized mouse group included two experiments: on the one hand, the amount of VLP in Exp. 1 was ~2,000 HA/mouse, and on the other hand, that in Exp. 2 was 1,000 HA/mouse. Red horizontal bar indicates the mean number of each group.

with membrane-bound IL-12 in a silkworm expression system (Maegawa et al, 2020). In preliminary tests, IL-12+HA-VLP exhibited protective activity against H1 and H5 influenza viruses. The level of protection against heterotypic virus, however, was lower-than-expected. Therefore, in this study, to enhance cross-reactivity against the different HA subtypes of the influenza A virus, we considered incorporating the M2 protein because it is highly conserved in all influenza A viruses and is implicated in viral protection. Our CCHA-VLP vaccines protected mice from the lethal infection of homologous and heterologous types of the influenza A virus, and suppressed viral replication in lung and protected infected mice from lung inflammation. Although single immunization with FkH5-VLP or CC-VLP inhibited lung inflammation, the survival rate of CC-VLP– or IL-12+FkH5-VLP–immunized mouse group in virus challenge was moderate, suggesting the incorporation of both CC and HA proteins may improve protective activity. Among CCIAH1-,

CCFkH5-, and CCAnH7-VLPs, only the CCFkH5-VLP–immunized mouse group showed a lower survival rate than the other two VLPs in the PRH1 challenge. This could be due to the fluctuation of experimental conditions in immunization and virus challenge (e.g., different amounts of immunized VLP, proportion of CC protein in CCHA-VLP sample, and susceptibility of individual ddY mouse). Although it may require some detailed investigation, such as comparison of VLP amount, as shown in the experiment with CCFkH5-VLP and BALB/c mouse, the CCHA-VLP showed broad protective activity against various serotypes of the influenza A virus. In particular, CCFkH5-VLP and CCAnAnH7-VLP showed protection against the heterogenous group of serotypes (Fig 10).

CCFkH5-VLP successfully induced not only homologous-type neutralizing antibody but also anti-M2 antibody and IFN-γ–producing cells, implying several possibilities on the mechanism of broad protective activity. Initially, IL-12 enhances the differentiation

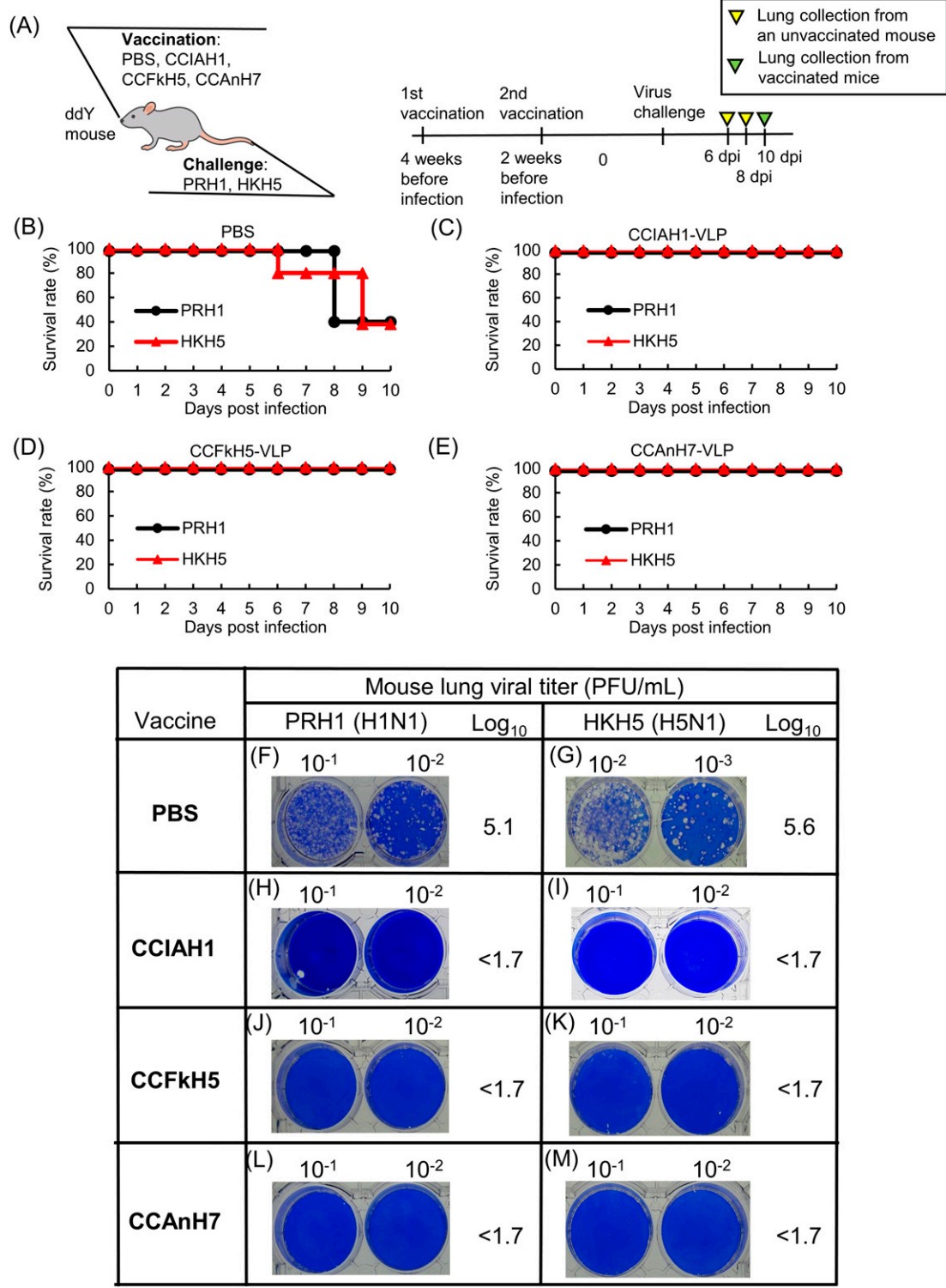

**Figure 5. Comparison of CCHA-VLP protective activity based on viral titers in mouse lungs.**
**(A)** Figure design shows mice vaccinated with three types of vaccine and subsequent challenge infection with PRH1 and HKH5 viruses (A). Simultaneously, the experimental week and date schedule are shown. **(B, C, D, E)** Five ddY mice in each group were immunized twice with PBS (B), CCIAH1-VLP (C), CCFkH5-VLP (D), and CCAnH7-VLP (E) at a 2-wk interval, and 2 wk after the last immunization, the mice were challenged with influenza viruses, PRH1 or HKH5. **(B, C, D, E)** Survival rates of each mouse group in lung collection experiments are shown (B, C, D, E). **(F, G, H, I, J, K, L, M)** Lower part of figure shows the results of viral titer in collected lungs (F, G, H, I, J, K, L, M). **(F, G, H, I, J, K, L, M)** Results of mice immunized with PBS (F, G), CCIAH1-VLP (H, I), CCH5-VLP (J, K), or CCH7-VLP (L, M) and infected with influenza viruses, PRH1 (F, H, J, L) or HKH5 (G, I, K, M), are shown. Lungs were collected from the challenge-infected mice. The samples of PBS-treated mice were obtained from dead mice, whereas the samples of immunized mice were collected at 10 dpi, the last day of observation. The collected lungs of each group were homogenized together, and the viral titers in the homogenates were determined by the plaque assay. The number of plaques was calculated in PFU/ml. The detection limit of this plaque assay was calculated as 50 ($1.7 \log_{10}$) PFU/ml.

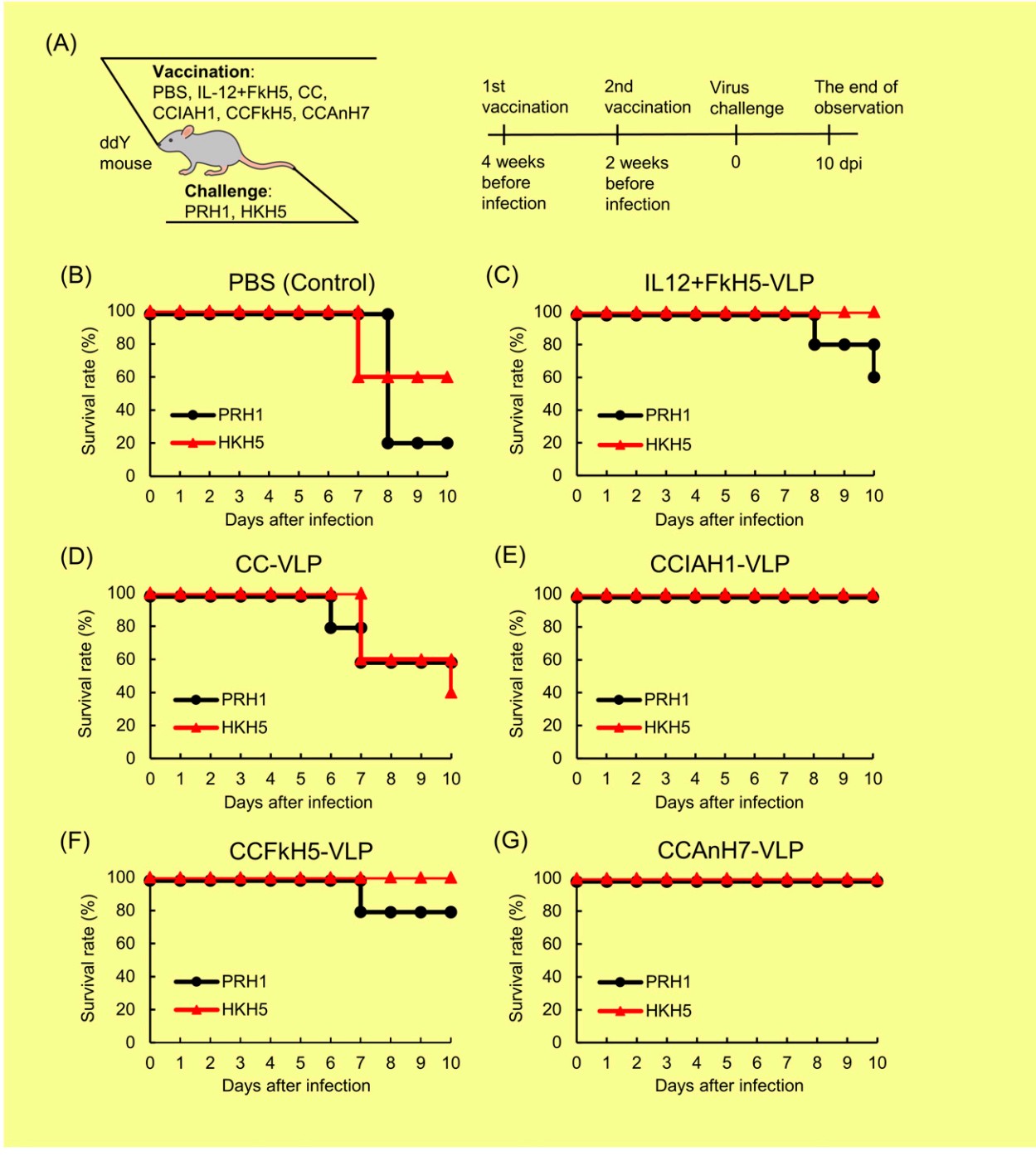

**Figure 6. Comparison of CCHA-VLP protective activity based on mouse survival rate.**
**(A)** Four types of vaccines and two challenge viruses around the mouse in addition to the experimental week and date schedule are shown (A). Five ddY mice in each group were immunized twice with PBS, IL-12+FkH5-VLP (558 μg HA protein/mouse), CC-VLP (558 μg CC protein/mouse), CCIAH1-VLP (67 μg HA protein/mouse), CCFkH5-VLP (270 μg HA protein/mouse), and CCAnH7-VLP (135 μg HA protein/mouse) at a 2-wk interval, and 2 wk after the last immunization, mice were challenged with PRH1 or HKH5 influenza viruses. **(B, C, D, E, F, G)** Survival rate of each group was determined within 10 dpi. Results are shown by each vaccine used: PBS (B), IL-12+FkH5-VLP (C), CC-VLP (D), CCIAH1-VLP (E), CCFkH5-VLP (F), or CCAnH7-VLP (G).

of CD4⁺ T cells to a Th1 phenotype and stimulates NK cells, leading to the production of an inflammatory cytokine (IFN-γ) (Metzger, 2010). In addition to the antiviral activity of IFN-γ, NK cells eliminate virus-infected cells independent from acquired immunity (Del Vecchio et al, 2007). Such IL-12–activated cellular immunity would be one of the contributed factors of broad protective activity. Although the response of IL-12 may be generally transient, we confirmed that the protective activity of IL-12–linked FkH5-VLP

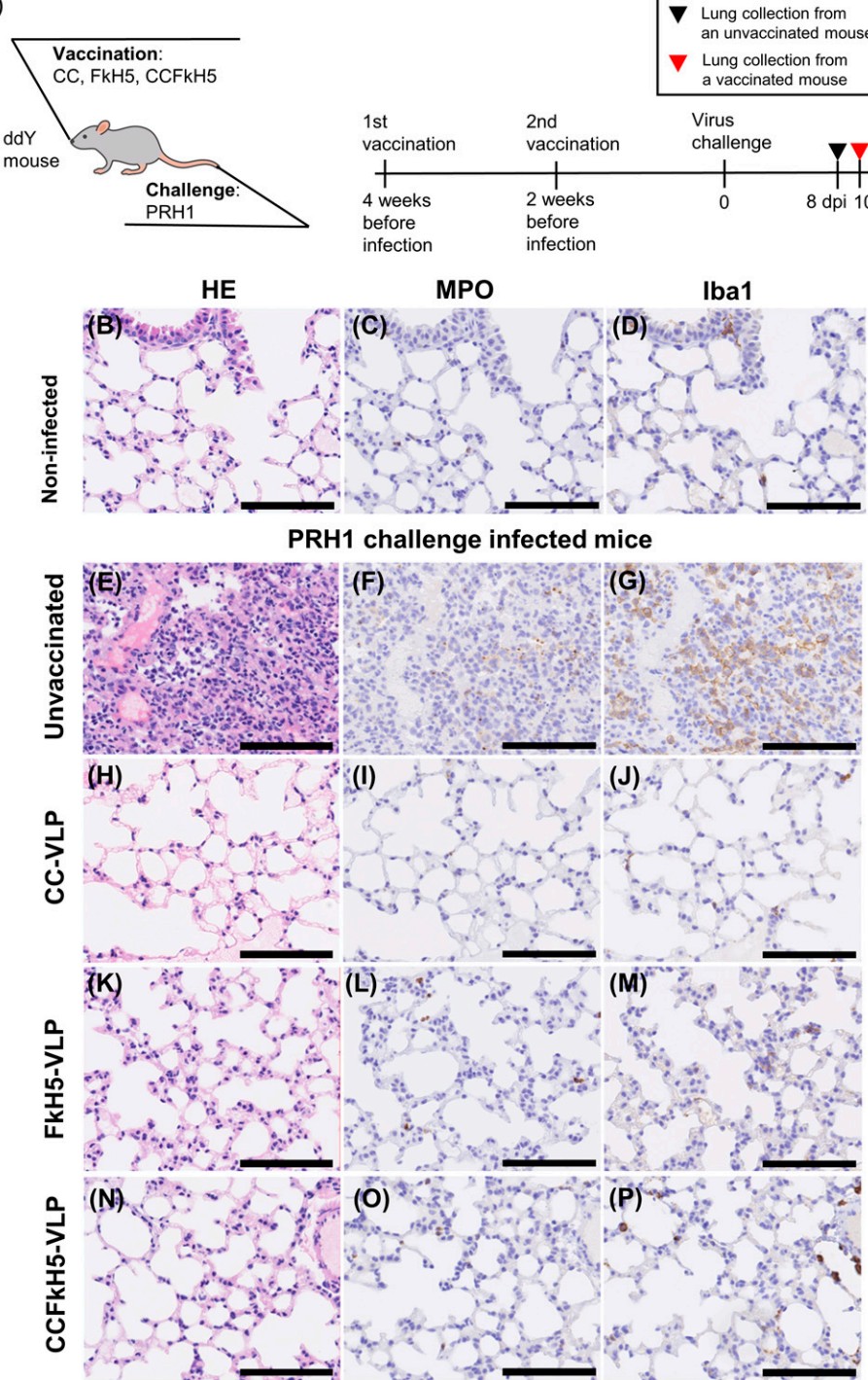

**Figure 7. Histopathological analysis of lung tissue from PRH1-infected mice.**
**(A)** Mice were immunized twice with CC-VLP, FkH5-VLP, and CCFkH5-VLP at a 2-wk interval, and 2 wk after the last immunization, mice were challenged with PRH1 influenza virus (A). **(B, C, D)** Lung from an unvaccinated and non-infected control mouse was also observed (B, C, D). **(E, F, G)** For the unvaccinated mouse, the lung was collected from a mouse that died at 8 dpi (E, F, G). **(H, I, J, K, L, M, N, O, P)** For the VLP-immunized mice, lungs were collected at 10 dpi (CC-VLP: (H, I, J); FkH5-VLP: (K, L, M); and CCFkH5-VLP: (N, O, P)). Collected lungs were fixed with 10% formalin for 1 d and subsequently embedded in paraffin. **(B, C, D, E, F, G, H, I, J, K, L, M, N, O, P)** Tissues were stained with hematoxylin and eosin (B, E, H, K, N), anti-MPO antibody (C, F, I, L, O), and anti-Iba1 antibody (D, G, J, M, P). MPO and Iba1 are markers of neutrophils and macrophages, respectively. Scale bar = 100 μm.

against heterotypic influenza A virus lasted until at least 6 wk after booster immunization as demonstrated in the previous study (Maegawa et al, 2020). The survival rate of the IL-12+FkH5-VLP–immunized mouse group in that experiment was the same as that in the present experiment. Naturally, humoral immunity would play an important role; for example, the M2 protein shows high homology among influenza A viruses and broad cross-protective activity

against a number of heterotypic influenza A virus strains (Kolpe et al, 2017). Similarly, HA2, the stalk of HA protein, also has high sequence similarity among influenza A viruses and can be a candidate of the epitope for broad immunity and moderate protective activity against several types of the influenza A virus induced by the HA2 vaccine (Adachi et al, 2019). From these series of evidence, the combination of IL-12, M2, and different HA antigens of the influenza

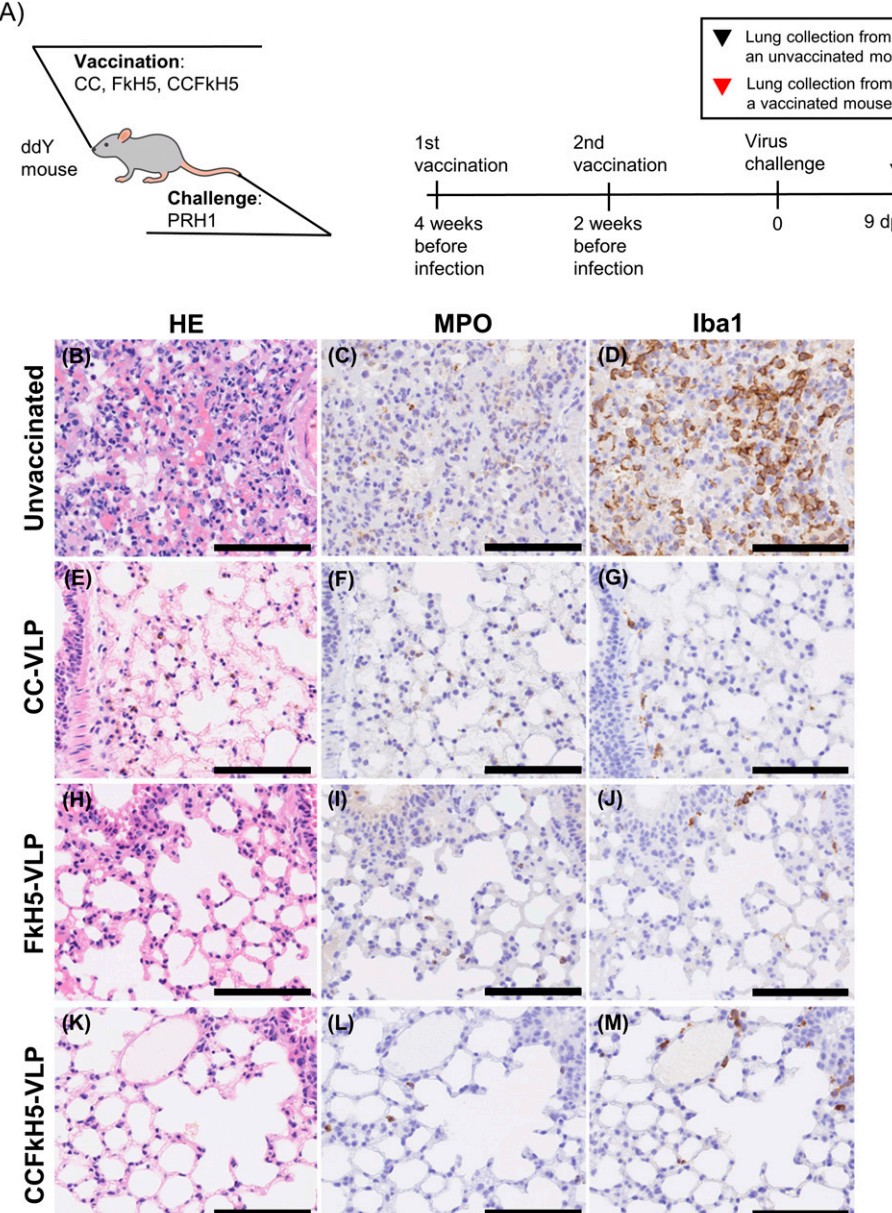

**(A)**

**Figure 8. Histopathological images of lung tissue from HKH5-infected mice.**
**(A)** Mice were immunized twice with CC-VLP, FkH5-VLP, or CCFkH5-VLP at 2-wk intervals, and 2 wk after the last immunization, mice were challenged with HKH5 influenza virus (A). **(B, C, D)** For the unvaccinated mouse, the lung was collected from a mouse that died at 9 dpi (B, C, D). **(E, F, G, H, I, J, K, L, M)** Samples of each VLP-immunized mouse were collected at 10 dpi (CC-VLP: (E, F, G); FkH5-VLP: (H, I, J); and CCFkH5-VLP: (K, L, M)). Collected lungs were fixed with 10% formalin for 1 d and subsequently embedded into paraffin. **(B, C, D, E, F, G, H, I, J, K, L, M)** Tissues were stained with hematoxylin and eosin (B, E, H, K), anti-MPO antibody (C, F, I, L), and anti-Iba1 antibody (D, G, J, M). MPO and Iba1 are markers of neutrophils and macrophages, respectively. Scale bar = 100 μm.

A virus may result in enhanced immune responses to surface antigens of the influenza A virus. As a result, the vaccination of CCHA VLP in mice has led to the production of effective antibodies M2, HA antigens, and cellular immunity. IL-12, which induces IFN-γ production and subsequent activation of B cells, also played an important role. Activated B cells enhance antibody responses against HA and M2 proteins, and probably led to broad cross-protective activity. From a series of the above evidence and the results obtained from the present study, the humoral and cellular immunity worked together and might have accomplished broad protective activity against various types of the influenza A virus.

This study had limitations. As mentioned above, vague and inconsistent experimental conditions in some parts of this study made the comparison and discussion of vaccine efficacy among VLPs difficult. These include an individual difference in the viral susceptibility of ddY mouse, the various amounts for immunization among VLPs, a relatively small number of samples for statistical analysis, and different sample collection dates between control and vaccinated mice. However, qualitative data including the successful induction of antibody and the inhibition of viral replication were obtained, and the results of infection experiments with BALB/c mice suggested that the CCFkH5-VLP vaccine showed significant protective activity against main influenza A virus subtypes (PRH1 and HKH5).

In conclusion, although there are several challenges to address, the CCHA-VLP vaccine showed successful broad protective activity against the influenza A virus. The vaccine can induce neutralizing activity to homologous virus, anti-M2 antibody, and cellular

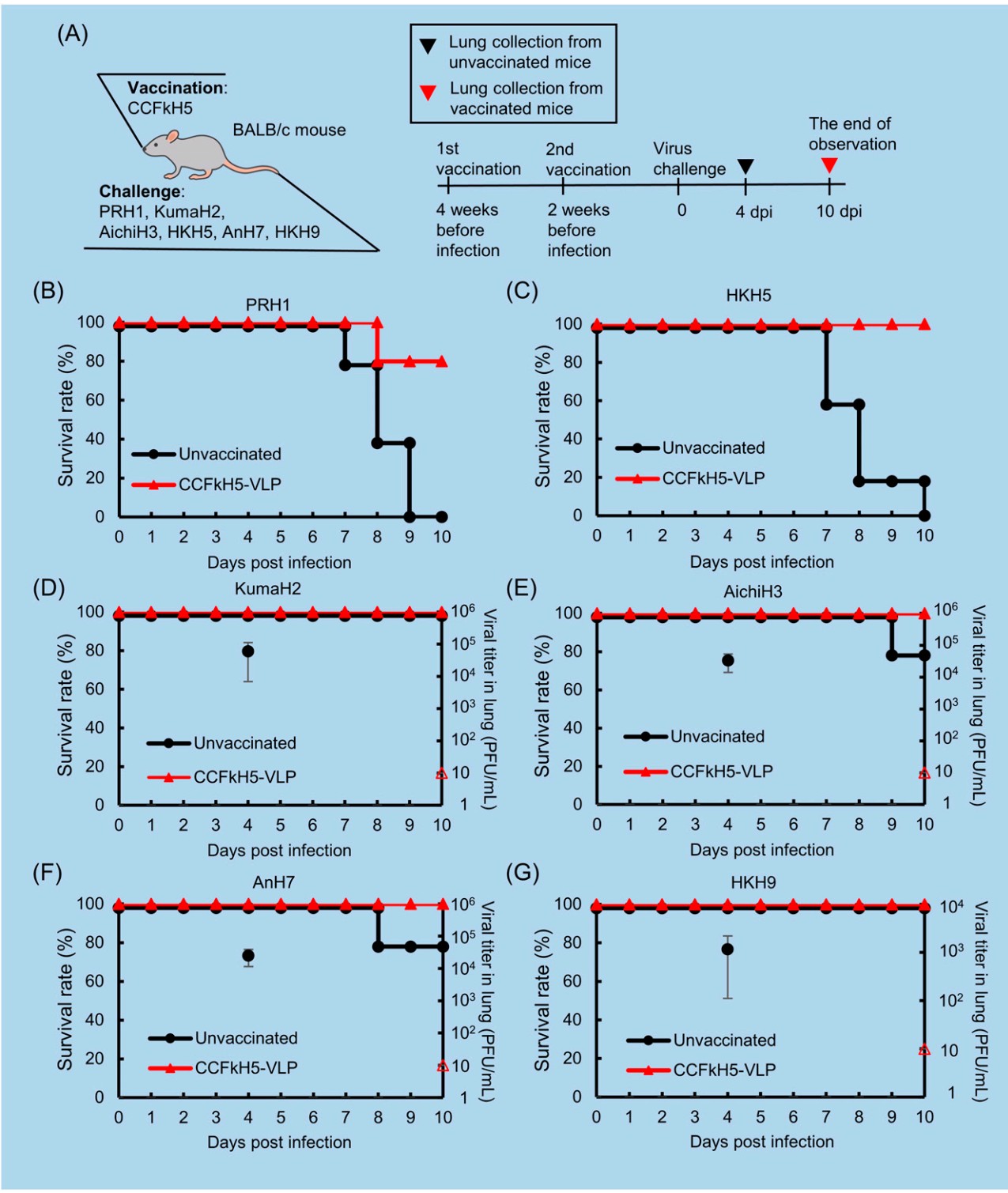

**Figure 9. Protective activity of CCFkH5HA-VLP vaccine against six subtypes of the influenza A virus in BALB/c mice.**
**(A)** shows the vaccine used in the protection test and subsequent infection with six subtypes of the influenza A virus around the mouse. **(A)** Simultaneously, the detailed experimental schedule is shown in (A). **(B, C, D, E, F, G)** BALB/c mice were immunized twice with CCFkH5-VLP (70 µg HA protein/mouse) at a 2-wk interval, and those mice were infected with PRH1 (B), KumaH2 (C), AichiH3 (D), HKH5 (E), AnH7 (F), and HKH9 (G) viruses. The survival rate of each group was observed for 10 d. Lethal viruses, PRH1, and HKH5 were inoculated with $5LD_{50}$/mouse. **(C, D, F, G)** Viral titer in the lung was measured for low-virulence viruses: KumaH2 ($7.6 \times 10^4$ PFU/mouse), AichiH3 ($1.1 \times 10^5$ PFU/mouse), AnH7 ($8.8 \times 10^4$ PFU/mouse), and HKH9 ($4.6 \times 10^4$ PFU/mouse) are shown with bar graphs (C, D, F, G). Lung was collected at 4 dpi for unvaccinated mice (black symbol), and at 10 dpi for vaccinated mice (red symbol). Open symbol means lower than the detection limit (10 PFU/ml).

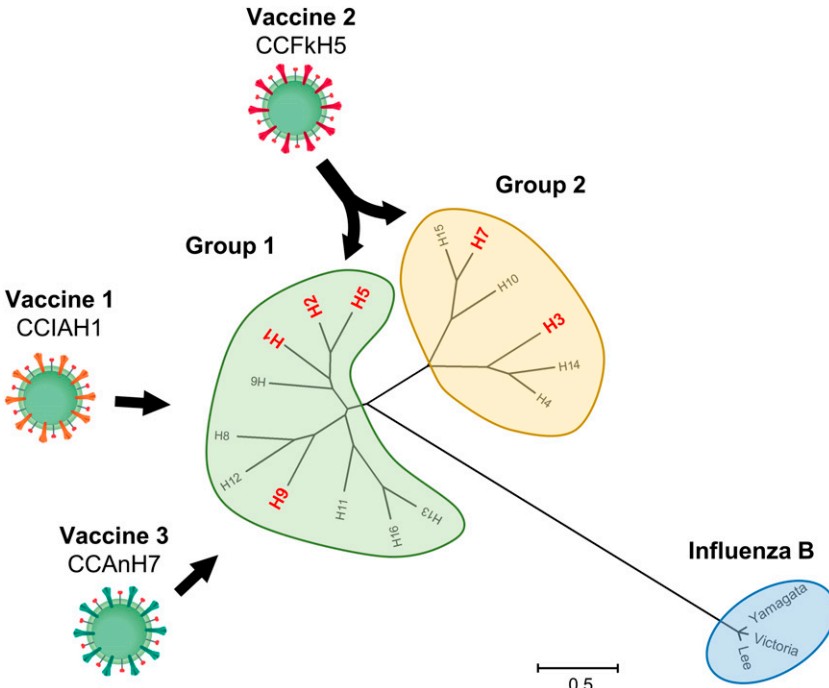

**Figure 10. Summary of the protection with the CCHA-VLP vaccine against various serotypes of the influenza A virus.** The protection with CCHA-VLP against various subtypes of the influenza A virus was summarized. Influenza A virus HA subtypes were further divided into groups 1 and 2 (Palese, 2020), and subtypes used in this study are indicated in red characters. Thick black arrows indicate that the vaccine showed protective activity against the HA group. The grouping of the influenza HA gene is shown with a maximum-likelihood phylogenetic tree. The substitution model, GTR+R+I, was used, and the tree was tested with 500 bootstrap replicates (bootstrap value is not shown). The length of the HA gene sequence used for calculation was 1,604 nucleotides.

immunity indicated by IFN-γ, supporting vaccine efficacy and mechanism. Although the present study finally reached the first stage of vaccine development, more effective synthetic immune modulators that are safe and possess wide immune reactivity must be developed in future research.

## Materials and Methods

### Design of CC

Based on the blueprint of IL-12, N2NA stalk, and M2 protein of influenza A H3N2 subtype, chimeric vaccines were synthesized by co-expressing HA protein with the mentioned three fusion proteins from their respective coding sequences. The entire structure of the CC molecule, consisting of IL-12 (cytokine), the stalk of NA, and M2 proteins, is shown in Fig 11A. To induce cross-protection and immune response against influenza VLP vaccines, we constructed a CC molecule, in order of N- to C-terminus, as follows: full-length M2 protein of A/Aichi/2/1968(H3N2) (ectodomain: 23 amino acids; transmembrane domain: 19 amino acids; and cytoplasmic domain: 54 amino acids), linker region (Gly–FLAG tag–8 amino acids–Gly–Gly), NA protein of A/Aichi/2/1968(H3N2) (cytoplasmic domain: 6 amino acids; transmembrane domain: 31 amino acids; and stalk region: 39 amino acids and 2 glycine linkers), and IL-12 (p40 subunit: 313 amino acids; linker: 19 amino acids; and p35 subunit: 193 amino acids), as depicted in Fig 11A. The stalk region of HA and M2e have previously been reported as candidates for universal vaccines; mice immunized with a peptide corresponding to the HA stalk (Wang et al, 2010) and a fusion protein of the M2 ectodomain with hepatitis B core protein (Neirynck et al, 1999) were challenged with divergent

subtypes of the influenza A virus. Based on these reports, we constructed a CC fusion gene but included the stalk region of NA rather than HA as we assumed that both NA and M2 ectodomains, albeit expressed as a fusion product, could be exposed to the extracellular space because they are type II and III transmembrane proteins, respectively. In addition, both proteins existed as tetramers. We also included IL-12 to explore its ability to enhance cytotoxic T lymphocyte and NK cell activation. Moreover, membrane anchor–type IL-12 has been reported to enhance the efficacy of influenza vaccines (Khan et al, 2014; Maegawa et al, 2020). The HA-AcNPV vectors containing Iowa H1N1 (IAH1) HA, Fukushima H5N1 (FkH5) HA, and Anhui H7N9 (AnH7) HA were infected into Eri silkworm pupae together with CC-AcNPV (Fig 11B). The three co-expressed CC fusion proteins along with HA on the surface of silkworm pupal cells formed the synthesized VLP (Fig 11C).

### Cells and viruses

The cells and viruses used in this study were described previously (Maegawa et al, 2018). Specifically, Madin-Darby canine kidney (MDCK) cells were grown in a MEM containing 5–10% fetal calf serum. Because highly pathogenic avian influenza H5 and H7 viruses were not available in this study because of the requirement of a biosafety level 3 (P3) facility, the low-virulence or avirulence viruses were used in the experiments investigating the vaccine protection. Influenza viruses were propagated in MDCK cells or 10-d-old embryonated chicken eggs. A/Puerto Rico/8/34 (H1N1; PRH1) was obtained from the Biomedical Science Association. A/swine/Iowa/15/1930 (H1N1; IAH1) was kindly provided by Prof. Y Sakoda, Faculty/School of Veterinary Medicine, Hokkaido University. A/Kumamoto/1/67 (H2N2; KumaH2) and A/Aichi/2/68 (AichiH3)

**(A)**

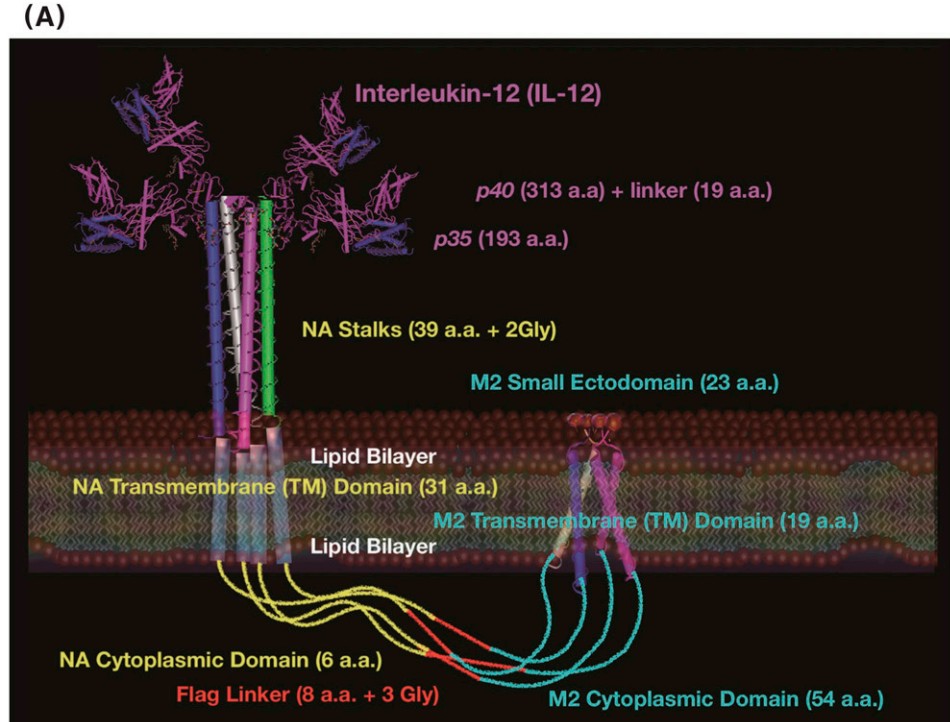

**Figure 11. Design of the CC molecule and production of the VLP vaccine.**
The concept of the CC molecule and CCHA-VLP was illustrated. **(A)** Visual representation of the entire CC molecule (A). CC comprises IL-12, headless influenza virus NA (stalk, transmembrane, and cytoplasmic domain of NA), and influenza virus matrix 2 protein (M2). The nucleotide and amino acid sequences of CC are shown in source data for Fig 2. **(B)** Recombinant AcNPV containing CC, IAH1 HA, FkH5 HA, and AnH7 HA genes was generated, and Eri silkworm pupae were co-infected with CC-AcNPV and each of IAH1HA-AcNPV, FkH5HA-AcNPV, or AnH7HA-AcNPV (B). **(C)** It was expected that the produced VLP would co-express the CC protein and each of the HA proteins on the surface (C).

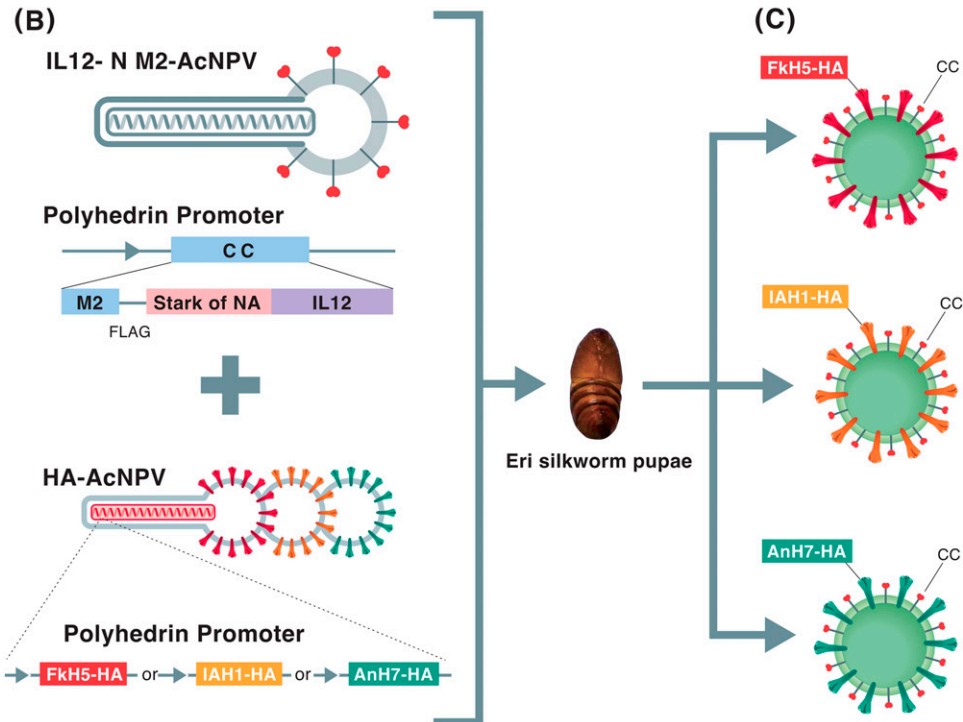

viruses were kindly provided by Dr. S Watanabe from the National Institute of Infectious Disease. Low pathogenic avian influenza (vaccine seed) viruses, RG-A/Barn Swallow/Hong Kong/1161/2010-A/PR/8/34 H5N1 [R] (6 + 2) (HKH5), and RG-A/Anhui/2013-A/PR/8/ 34 H7N9 [R] (6 + 2) (AnH7) were kindly provided by Dr. RG Webster from the St. Jude Children Research Hospital (Import Permit No.: 29 douken 322, issued on June 12, 2017, by the Ministry of Agriculture, Forestry and Fisheries). In addition to the above human, swine, and

avian strains, attenuated A/swine/Hong Kong/10/98 (H9N2) (HKH9) virus was also received from Dr. R Webster.

## Mouse experiments

4-wk-old female ddY and BALB/c mice (specific pathogen-free) were purchased from Japan SLC Co. Ltd. Testing the pathogenicity of the viruses used in the present study revealed that BALB/c mice tended to be more susceptible compared with ddY, and thus, they were used in the full-scale evaluation of the VLP vaccine.

## Ethics approval

Mouse experiments were performed in specific pathogen-free conditions in accordance with the Fundamental Rules for Animal Experiments and the Guidelines for Animal Experiments Performed at Nerome Institute of Biological Resources, published by the Animal Welfare and Animal Care Committee, including the Animal Ethics Committee of Nerome Institute of Biological Resources.

## Generation of recombinant AcNPV

The synthesis of influenza A HA genes for the A/tufted duck/Fukushima/16/2011 (FkH5), AnH7, swine Iowa H1N1 (IAH1), and CC fusion genes was performed by GENEWIZ. The synthesized genes were inserted downstream of the pFastBac polyhedron promoter, and the plasmids were transformed into DH10Bac-competent cells. Subsequently, the recombinant AcNPV DNA was purified and used to transfect Sf9 cells to generate recombinant AcNPV, as previously described (Maegawa et al, 2020, 2018).

## Production of VLP vaccines in silkworm pupae

Pupae of Eri silkworm were used to produce CC-VLP and CCHA-VLP vaccines. Eri silkworm pupae were infected with recombinant AcNPV coding for the CC and HA proteins, and the subsequent preparation of VLP vaccines was performed as previously described (Nerome et al, 2019). Briefly, the infected silkworm pupae were homogenized via six 2-min pulses of sonication on ice in PBS containing 0.01% formalin and phenylthiourea. The resultant homogenate was mixed with 20% (w/w) sucrose solution and further centrifuged at 25,000 rpm for 120 min using Optima LE-80K Ultracentrifuge with SW28 swing rotor (Beckman Coulter) at 4°C. Subsequently, CCHA-VLP samples were fractionated on continuous or discontinuous sucrose gradients consisting of 50% or 20% sucrose. The resultant VLP fraction formed on a 50% sucrose cushion was further purified by 50%, 40%, 30%, and 20% discontinuous sucrose density gradient centrifugations at 25,000 rpm for 120 min at 4°C, and each visible band was collected. For comparative analysis of HA fraction and its composition in VLP vaccines, the FkH5-AcNPV product was analyzed in detail. For example, a 10-fold dilution of original FkH5HA-VLP homogenate with PBS was treated with 40% saturated ammonium sulfate to identify and characterize the FkH5 VLP vaccine. Further purification methods were established using anion exchange chromatography and a fetuin column.

## Western blotting

Proteins were separated by sodium dodecyl sulfate–polyacrylamide gel electrophoresis on a 12.5% acrylamide gel. The separated proteins were transferred to polyvinylidene fluoride membranes (Clear Blot Membrane-P Plus; ATTO) using a semi-dry transfer system (EzFastBlot; ATTO), followed by blocking with BlockAce (KAC Co., Ltd.). After incubation with primary antibodies, the membrane was washed three times with PBS containing 0.1% Tween-20 (PBS-T) and incubated with HRP or alkaline phosphatase–conjugated secondary antibodies. After washing with PBS-T, the membrane was stained with Amersham ECL Select Western Blotting Detection Reagent (Cytiva), EzWestBlue (ATTO), or Immun-Star AP Substrate Pack (Bio-Rad).

## Immunofluorescence analysis

Sf9 cells were infected with recombinant AcNPVs and maintained in Grace's insect medium (Thermo Fisher Scientific) supplemented with 10% fetal bovine serum at 27°C for 1 d. Subsequent immunofluorescence was performed as previously described (Nerome et al, 2015).

## Antibodies

Anti-influenza A virus M2 monoclonal antibody (14C2; Abcam), anti-murine IL-12 (p70) polyclonal antibody (PeproTech), monoclonal anti-IL-12 antibody (Purified Rat Anti-Mouse IL-12 [p40/p70]; Becton, Dickinson and Company), and anti-HA (H1N1/California/06/2009) polyclonal antibody (Immune Technology Corp.) were purchased and used for Western blotting analysis and the FA assay. Antisera against swine IAH1, HKH5, and AnH7 viruses prepared from mice immunized with three doses (2,048–4,096 HA units) of purified viruses were also used for the detection of H5 or H7 HA proteins.

Antibodies against MPO (Agilent Technology) and Iba1 (Fujifilm Wako Pure Chemical Co.) were used for immunostaining for histological examination.

As secondary antibodies, goat anti-mouse IgG heavy and light chain cross-adsorbed antibody HRP-conjugated (Bethyl Laboratories, Inc.), goat anti-rabbit IgG H&L (Abcam), donkey anti-goat IgG H&L (HRP; Abcam), anti-mouse IgG (H + L) F(ab')2 fragment (Alexa Fluor 488 conjugate; Cell Signaling Technology), and donkey anti-goat IgG H&L (Alexa Fluor 555; Abcam) were used. Anti-N2 NA monoclonal antibody specific to the influenza A virus was purchased from Tokyo Chemical Industry Company. To identify the N2 NA, anti-Aichi/2/68 (H3N2) virus serum was also used in the examination.

## Hemadsorption test

Sf9 cells infected with CC-AcNPV, FkH5-AcNPV, or AnH7-AcNPV were washed with PBS, treated with 1% chicken erythrocytes, and incubated for 30 min at room temperature. The cells were then washed again with PBS and subjected to microscopy.

## TEM observation

The CCFkH5-VLP included in 50% and 20% sucrose cushion was further purified on discontinuous 50%, 40%, and 30% fractions at 25,000 rpm for 120 min at 4°C. The resultant CCFkH5-VLP, including 40% and 30% sucrose gradient, was diluted with PBS and centrifuged for 120 min at 25,000 rpm and 4°C. The pellet was resuspended in PBS and was used as a sample for electron micrography. The test samples were adsorbed on collodion-coated TEM grids (Veco, Cu, 400 mesh), negatively stained with 1% phosphotungstic acid (pH 7.0), and observed with a Tecnai G2 F20 electron microscope (FEI Company) operated at 120 kV.

## Immunization of mice with CC or CCHA-VLP vaccines and their challenge infection

Although a single radial diffusion test has been used for vaccine standardization in Japan and the UK, in this study, we employed the mass of HA protein converted from the previous saturated ammonium sulfate and anion exchange chromatography. Five groups of ddY mice (4 wk old, female) were intraperitoneally immunized with PBS, FkH5HA-VLP (558 µg HA protein/mouse), IL-12+FkH5 HA (CFkH5 HA)-VLP (558 µg HA protein/mouse), CC-VLP, CCIAH1HA-VLP (67 µg HA protein/mouse), CCFkH5HA-VLP (270 µg HA protein/mouse), and CCAnH7HA-VLP (135 µg HA protein/mouse) vaccines. Because IAH1 protein tended to show a minimal lethal toxicity to test mice, we administered a concentration of less than 100 µg/mouse for their safety. The former FkH5HA-VLP vaccine was used as a control vaccine without CC protein. As a result of the above condition, these seven groups of mice were immunized with different concentrations of VLP vaccines. However, the final evaluation of the CCFkH5HA-VLP vaccine was done in BALB/c mice at a dose of 70 µg HA protein/mouse. Booster immunization was done 2 wk later, and the immunized mice were infected with influenza viruses through the nasal route 4 wk post the first immunization. The ddY mice were infected with PRH1 ($7.9 × 10^4$ PFU/mouse), HKH5 ($1.6 × 10^5$ PFU/mouse), and AnH7 ($2.5 × 10^5$ PFU/mouse). Meanwhile, BALB/c mice were infected with PRH1 ($5LD_{50}$/mouse), KumaH2 ($7.6 × 10^4$ PFU/mouse), AichiH3($1.1 × 10^5$ PFU/mouse), HKH5 ($5LD_{50}$/mouse), AnH7 ($8.8 × 10^4$ PFU/mouse), and HKH9 ($4.6 × 10^4$ PFU/mouse). The survival rates of challenged mice were then observed for 10 d. Lung tissues were obtained from ddY mice infected with PRH1 and HKH5 and BALB/c mice infected with KumaH2, AichiH3, AnH7, and HKH9 for plaque assay. The lung of each mouse selected from several ddY groups was used for histological examination.

## Plaque formation assay

The lungs collected from mice were homogenized in PBS with a syringe and centrifuged at 4,000 rpm for 30 min using Beckman GS-15R with S4180 swing bucket roter (Beckman Coulter). The 10-fold serial diluted supernatants were prepared, and 200 µl was inoculated onto MDCK cells in a six-well plate. MEM containing 2.5 µg/ml of TPCK-trypsin and 0.9% agarose was overlaid in each well, and the cells were incubated for 3 d. After fixation with 10% formalin, the agarose gel was removed, and the cells were stained with Coomassie brilliant blue. The number of plaques was counted, and the PFU was calculated as the viral titer.

## HI test and plaque reduction test

Serum samples were collected from ddY mice tested in each challenge infection experiment at 10 d post-infection, the last day of observation. Sera from mice immunized with CCFkH5HA-VLP and CCAnH7HA-VLP and infected with PRH1, HKH5, and AnH7 were used for the HI test. Samples from the same group were pooled together and tested. The receptor-destroying enzyme–treated serum sample was twofold serial diluted with PBS in a 96-well plate and incubated with 16 HA of H1, H5, and H7 antigens for 30 min at room temperature. After incubation, it was mixed with 0.5% chicken red blood cells and incubated again for 60 min at room temperature, then observed.

Serum samples used for the plaque reduction test were prepared separately. The serum was collected from ddY mice twice immunized with CC-VLP, CCFkH5HA-VLP, or CCAnH7HA-VLP, but without infection. After receptor-destroying enzyme treatment, sera were 10-fold serial diluted, mixed with each PRH1, HKH5, AnH7, or HKH9 virus, and incubated for 30 min at room temperature. Each mixture was inoculated onto MDCK cells, and the plaque formation assay was performed. Neutralizing activity was calculated as the dilution rate of immunized serum at 50% plaque reduction compared with PBS.

## Histological examination

The lungs obtained from mice were fixed with 10% formalin for 1 d. Mice immunized with VLP and challenged with influenza viruses were euthanized on the last day of observation or on the first day that extreme body weight reduction was observed (<75% of initial body weight). In mice immunized with PBS and challenged with PRH1, a portion of the mice died without extreme reduction in their weight. The lungs of those mice were also collected. Their fixed lungs were paraffin-embedded, and subsequently deparaffinized in xylene and treated with 0.3% hydrogen peroxide in 100% methanol for more than 10 min to quench the endogenous peroxidase activity. Sections were heated at 95°C for 40 min in antigen retrieval solution, pH 9.0 (Nichirei Biosciences Inc.), followed by two rinses with PBS. Antibodies against MPO and Iba1 were used for immunostaining, which are neutrophil and macrophage markers, respectively. Histofine Simple Stain MAX PO(R) (Nichirei Biosciences Inc.) and Simple Stain DAB solution (Nichirei Biosciences Inc.) were used for staining. Tissues were also stained with hematoxylin and eosin at the Advanced Research Facilities and Services, Hamamatsu University School of Medicine.

## ELISA

Influenza M2e peptide solution was prepared in 5 µg/ml with 0.1 M carbonate–bicarbonate buffer (pH 9.5), and the solution was put in a flat-bottom microtitration plastic plate, then incubated overnight at 4°C. M2e peptide solution was removed and washed with PBS containing 0.05% Tween-20, and blocked with four times dilution of BlockAce solution for 30 min at room temperature. After washing, the tested sera were added to wells and incubated for 2 h at room temperature. After washing, the HRP conjugate goat anti-mouse IgG

antibody (1:10,000) was added and incubated for 1 h at room temperature. After washing, a one-step TMB ELISA substrate (Thermo Fisher Scientific) was used for detection according to the manufacturer's protocol.

### Mouse IFN-γ ELISpot assay

10-wk-old BALB/c mice were immunized with 70 μg HA protein/mouse of the CCFkH5HA-VLP vaccine twice at an interval of 2 wk, and their spleens were collected 2 wk after the second immunization. Non-immunized mice of the same age were also used as the unvaccinated control group. Collected spleens were homogenized in a serum-free medium, and the suspension was collected and incubated on ice until the heavier fraction settled down. The supernatant was collected and centrifuged at 300$g$ for 5 min, and the supernatant was removed. The pellet was resuspended in 10 ml serum-free medium and centrifuged at 300$g$ for 5 min again. The resulting pellet was resuspended in 1 ml of 0.83% $NaNH_4$ for hemolysis and left for 5 min at room temperature. The suspension was mixed with 10 ml serum-free medium and centrifuged at 300$g$ for 5 min followed by removing the supernatant, followed by the same wash step once. The pellet was resuspended in 5 ml serum-free medium and applied through a Falcon 40-μm cell strainer (Corning). The number of cells was adjusted to $10^6$–$10^7$ cells/ml and stored at −80°C until further use. Collected mouse splenocytes were added at $10^5$ cells/well and stimulated with 16 HA titers of the H5N1 virus for 24 h. IFN-γ–secreting cells were detected using mouse IFN-γ Single-Color ELISpot (Cellular Technology Limited), according to the manufacturer's protocol. Spots were stained, counted, and calculated as the number of IFN-γ–secreting cells per million splenocytes.

### Constructing the phylogenetic tree of the HA protein of the influenza A virus

Subtypes of the influenza A virus HA protein were classified using a phylogenetic tree. Specifically, a maximum likelihood tree was constructed using the Molecular Evolutionary Genetics Analysis X program with general time reversible + G + I as a substitution model. The reliability of the tree was tested with 500 bootstrap replicates. Subtypes of the HA gene were classified into groups 1 and 2 according to a reference in Fields Virology (Palese, 2020). The nucleotide sequence of each HA subtype was retrieved from GenBank, and the accession numbers of those sequences are shown in Table S1.

### Statistical analysis

Statistical analysis to compare the means of IFN-γ–producing cells was performed by Welch's test using the R software version 3.0.0 (R Core Team, 2019). $P$-value less than 0.05 was considered as statistically significant. The statistical method used and the number of data replicates are shown in the corresponding figure legend.

## Supplementary Information

## Acknowledgements

The Eri silkworms used in this study were donated by Shinshu University according to a Grant-in-Aid "National Bio Resource Project (NBRP, RR2002), Silkworm Genetic Resources" for Scientific Research from the Ministry of Education, Science, Sports and Culture of Japan. Moreover, we thank Ms. M Yonamine and Ms. M Uehara from Nago Commercial & Technical High School for their help with figure preparation. This research did not receive any specific grant from the public, commercial, or not-for-profit funding sectors.

### Author Contributions

K Nerome: conceptualization, supervision, funding acquisition, methodology, project administration, and writing—original draft, review, and editing.
T Imagawa: data curation, formal analysis, investigation, visualization, and writing—review and editing.
S Sugita: conceptualization and visualization.
Y Arasaki: investigation.
K Maegawa: resources and investigation.
K Kawasaki: investigation.
T Tanaka: investigation.
S Watanabe: resources.
H Nishimura: resources.
T Suzuki: methodology.
K Kuroda: resources.
I Kosugi: methodology.
Z Kajiura: resources.

### Conflict of Interest Statement

The authors declare that they have no conflict of interest.

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
