## [Reviewer comments · Life Science Alliance]

Life Science Alliance

The potential of a universal influenza virus-like particle vaccine expressing a chimeric cytokine

Kuniaki Nerome, Toshifumi Imagawa, Shigeo Sugita, Youta Arasaki, Kenichi Maegawa, Kazunori Kawasaki, Tsuyoshi Tanaka, Shinji Watanabe, Hidekazu Nishimura, Tetsuro Suzuki, Kazumichi Kuroda, Isao Kosugi and Zenta Kajiura
DOI: <https://doi.org/10.26508/lsa.202201548>

Corresponding author(s): Dr. Kuniaki Nerome (Nerome Institute of Biological Resources); Toshifumi Imagawa

Review Timeline:

Submission Date:	2022-06-07
Editorial Decision:	2022-07-25
Revision Received:	2022-10-16
Editorial Decision:	2022-10-17
Revision Received:	2022-10-19
Accepted:	2022-10-21

Transaction Report:

July 25, 2022

Re: Life Science Alliance manuscript #LSA-2022-01548-T

Dr. Kuniaki Nerome
Nerome Institute of Biological Resources
893-2
Nakayama
Nago, Okinawa 905-0004
Japan

Dear Dr. Nerome,

Thank you for submitting your manuscript entitled "The potential of a universal influenza virus-like particle vaccine expressing a chimeric cytokine" to Life Science Alliance. The manuscript was assessed by expert reviewers, whose comments are appended to this letter. We invite you to submit a revised manuscript addressing the Reviewer comments.

Thank you for this interesting contribution to Life Science Alliance. We are looking forward to receiving your revised manuscript.

Sincerely,

B. MANUSCRIPT ORGANIZATION AND FORMATTING:

Reviewer #1 (Comments to the Authors (Required)):

In this manuscript by Nerome et al the authors describe a VLP vaccine they constructed that contains a fusion protein "CC" of IL12, NA and M2, combined with HA1, HA5 and HA7, which are produced in silkworm cells. The CC-HA VLP vaccination in mice were shown to perform better than the non-IL12 containing counterpart in protecting against influenza disease for multiple HA subtypes. The authors suggest the chimeric cytokine VLP vaccine may form a foundation of a universal influenza vaccine.

This manuscript contains relatively straightforward data and the figures are for the most part clearly presented. Overall, it appears that inclusion of the CC fusion protein with IL-12 and M2 in VLPs has some beneficial effect on protection, although the mechanism is unclear. The use of IL-12 may activate cellular immunity such as NK cells but one wonders why such an approach has not seen much uptake in the field despite being reported in 2014 by Kahn et al as the authors have cited. One concern is whether the effect of IL-12 is transient, especially since viral challenge was only 2 weeks after immunizations. This reviewer is concerned about vaccine durability using this approach, since the effect of IL-12 would wane over time, in which case humoral immunity may be the best correlate of protection that may be independent of the effect of IL-12. Unfortunately, the authors did not measure flu neutralizing antibody titers of the vaccines, nor did they measure the titers to M2 vs HA, so the mechanism underlying the difference in protection due to IL-15 is unclear as is the durability of the effect. The authors should address durability and mechanism of protection in the manuscript. One final note, there are a lot of constructs with difficult names making it challenging to follow what is going on in the manuscript.

Specific comments

p.6. In 103. "...vaccine immunogenicity, IL-12 must be incorporated" The authors have not demonstrated necessity of including IL-12 in a flu vaccine. While inclusion of IL-12 may have shown benefit, many other adjuvants and methods also show benefits. The authors should therefore change "must" in this sentence to "might provide benefit by being" or tone down the sentence in some equivalent manner.

p.7. In 122. HA5 does not seem to be co-expressed in majority of cells examined in Fig.1 while M2, IL15 and HA7 are. This should be stated in the text. Can the authors please also comment on why this might be?

p.8.138-143. The first sentence should be deleted as it states the conclusion before the results. Western blotting shows a 100 kDa band, but nowhere is it shown what size band to expect, so there should be a predicted MW and sequence shown of the construct. Then the authors state the HA proteins were verified by immunofluorescence; however, those panels look like more Western blot results rather than immunofluorescence. Please explain, clarify.

p.8 Bottom section. The authors refer to Fig.3 but do not introduce the method of analysis. They should introduce the kind of ? negative stain EM they are using.

p.9 In 157-166. The authors state there are hybrid or recombinant particles. This is confusing without explanation. How can the authors tell by looking at the particles whether they are recombinants or not, and what are their morphological criteria for determining such a particle? The authors state they are concerned about the presence of recombinant particles. The authors must state what are their specific concerns?

p.10 Top section. This whole section is based on a Supplementary Table. These data should be in the main manuscript if they are to be the subject of a whole section in the main body text.

p.11,12. Ln 198-210. In comparing Fig.5C to Fig.5G it appears the survival from PR1 infection increases from 60% to 80% due to IL-15 fusion, which is only a modest increase, compared to the 100% survival with the other CC-HA vaccine constructs. Can the authors offer a plausible explanation for this result?

p.12. 212- . The authors describe the histological staining with HE stain, MPO and Iba1 stains. The authors did not state which cell features and cell types each stain is being used for in their investigation, but they should state these points at the beginning

of the section.

p.15. In 274. "This result was accomplished by combining the fusion protein with HA." The authors should state the precise name and components of the fusion protein.

p.16. "Cytokine treatment has been reported as an effective strategy against several communicable diseases " The authors cite the Khan paper from 2014 that used IL-12/IL-23 to adjuvant a whole inactivated flu vaccine. However, it does not appear to be a highly cited paper and similar IL-12 cytokine molecular adjuvants have not been much reported since. One wonders whether this approach is that effective relative to the many other approaches to flu vaccines in the field. Moreover, the effect of IL-15 may be transient, so the real question to be addressed is whether vaccine efficacy is durable beyond 2 weeks to say, a couple of months?

p.18. In 324. "Third, data analysis was not exact due to inconsistent data." This is a troubling statement. The authors must be more precise and concrete here, and state exactly which data was inconsistent and by how much was the inconsistency?

p.18. In 329. "...with an advantage to alter the combination by changing the CC protein" This phrase was not understood by this reviewer. The authors should clarify exactly what is meant by this phrase.

p.19. The authors describe in the text their "CC" fusion protein; for clarity, they should include a figure showing the complete, annotated amino acid sequence.

Reviewer #2 (Comments to the Authors (Required)):

the manuscript entitled "The potential of a universal influenza virus-like particle vaccine expressing a chimeric cytokine" by Kuniaki Nerome produced virus-like particle vaccines, expressing a chimeric cytokine, and demonstrates the protective efficacy against different strains of the influenza virus. The results of this study can lay the foundation for designing the universal influenza vaccine that protects against future pandemics.

However, the following improvements have to be done before it can be considered for publication.

1.The evaluation of protective efficacy of the CCHA-VLP vaccines was based on the survival rate of mice and the viral titers in the target lung, it is necessary to evaluate the production of cytokines and the primed NK cells or T cell response.

2.In page 11, line 201,"PBS-immunized mice challenged with PRH1 or HKH5 viruses showed 20% and 40% survival, respectively (Fig. 5B)", the survival rate was inconsistent with that in Fig. 5B. please check and confirm it.

Response to comments from Reviewer 1

To address the reviewer's comments, we performed additional experiments. We added data on induction of neutralizing activity and T cell response by CCHA-VLP vaccine, and persistence of vaccine efficacy. We have mentioned the mechanism of our VLP vaccine in the Discussion section. Moreover, the names of VLPs were simplified to be more comprehensive. For example to easily distinguish CCFkH5-VLP from CFkH5-VLP, "CFkH5-VLP" was changed to IL-12+FkH5-VLP.

Page 16, lines 283–285

CCFkH5-VLP successfully induced not only homologous typic neutralizing antibody, but also anti-M2 antibody and IFN- γ producing cells, implying several possibilities on the mechanism of broad protective activity.

p.6. In 103. "...vaccine immunogenicity, IL-12 must be incorporated" The authors have not demonstrated necessity of including IL-12 in a flu vaccine. While inclusion of IL-12 may have shown benefit, many other adjuvants and methods also show benefits. The authors should therefore change "must" in this sentence to "might provide benefit by being" or tone down the sentence in some equivalent manner.

Response: Thank you for the valuable comment. We have revised the following sections of the manuscript accordingly.

Pages 5–6, lines 89–90

In our previous study, to enhance vaccine immunogenicity based on above evidences of cytokine usability, IL-12 was incorporated into influenza HA vaccine.

Page 5, lines 83-88 (Evidence of cytokine usability)

For instance, natural killer (NK) cells are activated by interferon (IFN)- α , - β , and - γ , and interleukin (IL)-12, -15, and -18. Among these, IL-12 and IL-18 specifically activate NK cells (Kobayashi et al., 1989). NK cells and IFN- γ enhancing B cell responses are important for antiviral activity. Previous studies have reported that cytokine treatment is an effective regimen against several communicable diseases (Khan et al., 2014; Jiang et al., 1999; Fujioka et al., 1999; Gai et al., 2017).

p.7. In 122. HA5 does not seem to be co-expressed in majority of cells examined in

Fig.1 while M2, IL15 and HA7 are. This should be stated in the text. Can the authors please also comment on why this might be?

Response: Thank you for the comment. HA5 not being co-expressed in majority of the cells could be due to the fact that we did not consider the titer of each AcNPV recombinant. Therefore, the number of infected cells in each AcNPV was different. We acknowledge this as an important information and added this information to our revised manuscript:

Pages 6-7, line 109-111

Next, the CC-AcNPV + FkH5-AcNPV infected cells and the CC-AcNPV + AnH7-AcNPV infected cells were prepared and co-expression of HA and IL12 was partially detected (Fig. 1D and E).

Page 14, line 253-257

Next, the result of immune fluorescence analysis with sf9 cells showed that the single infection of either CC-AcNPV or HA-AcNPV was not negligible in the cells co-infected with CC-AcNPV and HA-AcNPV. This may be because the titer of each AcNPV was not considered in this experiment, and thus the number of infected cells in each AcNPV differs.

p.8.138-143. The first sentence should be deleted as it states the conclusion before the results. Western blotting shows a 100 kDa band, but nowhere is it shown what size band to expect, so there should be a predicted MW and sequence shown of the construct. Then the authors state the HA proteins were verified by immunofluorescence; however, those panels look like more Western blot results rather than immunofluorescence. Please explain, clarify.

Response: Thank you for the valuable comment. The first sentence “The CC protein was successfully expressed in CC-AcNPV infected silkworm pupae.” was deleted. The expected molecular size was shown in the legend of Figure 2, and the discrepancy was mentioned in the Discussion section. The nucleotide sequence of chimeric cytokine was attached as a supplementary material.

Page 14, line 250-253

Regarding the expression of VLPs, western blot of CC showed a band over 100 kDa in

molecular size, although the calculated molecular size of CC was approximately 79.6 kDa. This discrepancy was probably due to the sugar chain linkage, as it is estimated that the amino acid sequence of CC includes ten N-glycans binding sequences.

p.8 Bottom section. The authors refer to Fig.3 but do not introduce the method of analysis. They should introduce the kind of negative stain EM they are using.

Response: Following the Reviewer's suggestion, the method of TEM was described in the Results section.

Pages 7–8, lines 128–129

“Purified VLP samples, including FkH5-, CC- and CCFkH5-VLPs, were negatively stained with 1% phosphotungstic acid (pH 7.0) and observed with transmission electron microscopy (TEM).”

p.9 In 157-166. The authors state there are hybrid or recombinant particles. This is confusing without explanation. How can the authors tell by looking at the particles whether they are recombinants or not, and what are their morphological criteria for determining such a particle? The authors state they are concerned about the presence of recombinant particles. The authors must state what are their specific concerns?

Response: Thank you for the valuable comment. We used the word “recombinant” for baculovirus AcNPV, such as CC-AcNPV and FkH5-AcNPV. In contrast, “hybrid” means the VLP containing CC and FkH5 HA proteins. To simplify, the word “recombinant” was removed in the part of “Morphological analysis of CCHA-VLP structure” in the revised manuscript. Regarding identifying VLPs, we distinguished CCFkH5-VLP (hybrid) and CC-VLP or FkH5-VLP (containing single protein) through TEM observation. Comparing whether the length of projections were similar to that of the influenza virus, from the observation of FkH5-VLP and CC-VLP, the length of projection of these VLP containing single protein was different from each other.

We were concerned on the proportion of hybrid CCFkH5-VLP. It could be that low or high proportion of hybrid CCFkH5-VLP (e.g. <20% or >80%), might affect vaccine efficacy. However, since that was not the main objective in the manuscript, it will be topic for future research. We have indicated this in the revised manuscript.

Page 14, 260-261:

However, it is unclear whether co-existence of CC and HA on the surface of same VLP particle is important for immunogenicity or not, and thus should be the focus of future studies.

p.10 Top section. This whole section is based on a Supplementary Table. These data should be in the main manuscript if they are to be the subject of a whole section in the main body text.

Response: Thank you for your suggestion. The information on Supplementary Table 1 was incorporated in Table 1 of the revised manuscript.

p.11,12. Ln 198-210. In comparing Fig.5C to Fig.5G it appears the survival from PRH1 infection increases from 60% to 80% due to IL-12 fusion, which is only a modest increase, compared to the 100% survival with the other CC-HA vaccine constructs. Can the authors offer a plausible explanation for this result?

Response: Thank you for your comment. There are several possibilities. First, it could be due to the susceptibility of ddY mouse against influenza virus. It could be that the ddY mouse was less susceptible against influenza virus than BALB/c mouse, and ddY (outbred) mouse has higher genetic diversity than BALB/c (inbred) mouse. In reality, the mortality of infection experiment with ddY mouse was sometimes unstable although the same lot of inoculum was used. The individual showing relatively high susceptibility might be inadvertently included in the CCFkH5-VLP vaccinated group, and probably triggered the difference of protective activity among three CCHA-VLP vaccines. Moreover, the amount of immunized VLP was not measured exactly. The amount was estimated from HA titer, without considering the amount of CC protein. Therefore, it was difficult to estimate the vaccine efficacy especially based on CC protein.

Consequently, the comparison of efficacy among VLP vaccines was not mentioned, and only the qualitative results were discussed.

p.12. 212- . The authors describe the histological staining with HE stain, MPO and Iba1 stains. The authors did not state which cell features and cell types each stain is being used for in their investigation, but they should state these points at the beginning of the section.

Response: Thank you for the comment, and we apologize for being unclear. The cell features and cell types used was indicated in the histological test result.

Page 12, lines 214-215

“MPO and Iba1 were markers of neutrophil and macrophage respectively, and were used to indicate signs of inflammatory.”

p.15. In 274. "This result was accomplished by combining the fusion protein with HA." The authors should state the precise name and components of the fusion protein.

Response: The relevant statement has been removed as we have re-written the Discussion section.

p.16. "Cytokine treatment has been reported as an effective strategy against several communicable diseases <refs>" The authors cite the Khan paper from 2014 that used IL-12/IL-23 to adjuvant a whole inactivated flu vaccine. However, it does not appear to be a highly cited paper and similar IL-12 cytokine molecular adjuvants have not been much reported since. One wonders whether this approach is that effective relative to the many other approaches to flu vaccines in the field. Moreover, the effect of IL-12 may be transient, so the real question to be addressed is whether vaccine efficacy is durable beyond 2 weeks to say, a couple of months?

Response: Thank you for your comment. In this study, the data two weeks after IL12+FkH5-VLP booster immunization was included and 60% of immunized mice survived in the PRH1 virus challenge. In fact, in previous study, we tested the vaccine efficacy six weeks after the IL-12+FkH5-VLP booster immunization with PRH1 virus challenge; the survival rate was also 60%. Although the amount of immunized VLP was different between two experiments, it was observed that the protective activity against heterotypic virus was sustained for at least six weeks. From this result, IL-12 containing HA VLP vaccine has prolonged protectivity compared with the transient effect of IL-12 as generally thought. We mentioned this information in the Discussion section.

Page 14, line 289-293

Although response of IL-12 may be generally transient, we confirmed that the protective activity of IL-12 linked FkH5-VLP against heterotypic influenza A virus lasted until at least six weeks after booster immunization as demonstrated in previous

study (Maegawa et al., 2020). The survival rate of IL-12+FkH5-VLP immunized mouse group in that experiment was same to that in the present experiment.

p.18. In 324. "Third, data analysis was not exact due to inconsistent data." This is a troubling statement. The authors must be more precise and concrete here, and state exactly which data was inconsistent and by how much was the inconsistency?

Response: Thank you for the valuable comment. We apologize for being unclear. The data inconsistency includes the different sampling timing and different criteria for sample collection, making the data analysis difficult. As this is a similar statement to that of the first limitation, we have revised this section as follows:

Page 17, line 307–311

This study had limitations. As mentioned above, vague and inconsistent experimental conditions in some parts of this study, made the comparison and discussion of vaccine efficacy among VLPs, difficult. These include individual difference in viral susceptibility of ddY mouse, the various amount for immunization among VLPs, relatively small number of sample for statistical analysis, and different sample collection date between control and vaccinated mice.

p.18. In 329. "...with an advantage to alter the combination by changing the CC protein" This phrase was not understood by this reviewer. The authors should clarify exactly what is meant by this phrase.

Response: Thank you for your comment. The last part of the Discussion section was rewritten and the relevant statement was deleted in the revised manuscript.

p.19. The authors describe in the text their "CC" fusion protein; for clarity, they should include a figure showing the complete, annotated amino acid sequence.

Response: Thank you for the suggestion. The figure showing the nucleotide and amino acid sequences of CC fusion protein was included as a supplementary information.

Response to comments from Reviewer 2

The evaluation of protective efficacy of the CCHA-VLP vaccines was based on the survival rate of mice and the viral titers in the target lung, it is necessary to evaluate the

production of cytokines and the primed NK cells or T cell response.

Response: Thank you for the comment. The manuscript has included the result of mouse IFN- γ ELISPOT assay with CCFkH5 VLP vaccinated mice in Fig. 4. Since NK cell and T cell produce IFN- γ , response of those cell was evaluated by induction of IFN- γ producing cells.

In page 11, line 201, "PBS-immunized mice challenged with PRH1 or HKH5 viruses showed 20% and 40% survival, respectively (Fig. 5B)", the survival rate was inconsistent with that in Fig. 5B. please check and confirm it.

Response: Thank you for the valuable suggestion. According to reviewer's comment, the statement in the manuscript was matched to that in Figure 6B (Fig. 5B changed to Fig. 6B, because of insertion of Fig. 4).

Page 11, lines 199-200

PBS-immunized mice challenged with PRH1 or HKH5 viruses showed 20% and 60% survival, respectively (Fig. 6B).

October 17, 2022

RE: Life Science Alliance Manuscript #LSA-2022-01548-TR

Dr. Kuniaki Nerome
Nerome Institute of Biological Resources
893-2
Nakayama
Nago, Okinawa 905-0004
Japan

Dear Dr. Nerome,

Thank you for submitting your revised manuscript entitled "The potential of a universal influenza virus-like particle vaccine expressing a chimeric cytokine". We would be happy to publish your paper in Life Science Alliance pending final revisions necessary to meet our formatting guidelines.

- please add ORCID ID for first corresponding author-you should have received instructions on how to do so
- please add the Twitter handle of your host institute/organization as well as your own or/and one of the authors in our system
- please recheck your figure callouts for Figure 10 and remove the panel A from the callout because this is not in the legend (because it is the only panel in the figure, you do not have to designate it with a letter)
- please add a callout for Figure 8A, 8H, and 8K to your manuscript
- please remove the section entitled "Online supplementary materials"

Figure Check:

- Figure S1 should instead be marked as a source data file "Source Data for Figure 2"

A. FINAL FILES:

B. MANUSCRIPT ORGANIZATION AND FORMATTING:

Sincerely,

October 21, 2022

RE: Life Science Alliance Manuscript #LSA-2022-01548-TRR

Dr. Kuniaki Nerome
Nerome Institute of Biological Resources
893-2
Nakayama
Nago, Okinawa 905-0004
Japan

Dear Dr. Nerome,

Thank you for submitting your Research Article entitled "The potential of a universal influenza virus-like particle vaccine expressing a chimeric cytokine". It is a pleasure to let you know that your manuscript is now accepted for publication in Life Science Alliance. Congratulations on this interesting work.

DISTRIBUTION OF MATERIALS:

Again, congratulations on a very nice paper. I hope you found the review process to be constructive and are pleased with how the manuscript was handled editorially. We look forward to future exciting submissions from your lab.

Sincerely,
